# Isoprene derived secondary organic aerosol in the global aerosol-chemistry-climate model ECHAM6.3.0-HAM2.3-MOZ1.0

Scarlet Stadtler[1], Thomas Kühn[2,3], Sabine Schröder[1], Domenico Taraborrelli[1], Martin G. Schultz[1,*], and Harri Kokkola[2]

[1]Institut für Energie- und Klimaforschung, IEK-8, Forschungszentrum Jülich, Germany
[2]Finnish Meteorological Institute, P.O. Box 1627, 70211, Kuopio, Finland
[3]Department of Applied Physics, University of Eastern Finland, P.O. Box 1627, 70211, Kuopio, Finland
[*]Now at Jülich Supercomputing Centre, JSC, Forschungszentrum Jülich, Germany

*Correspondence to:* Harri Kokkola (harri.kokkola@fmi.fi)

## Abstract

Within the framework of the global chemistry climate model ECHAM-HAMMOZ a novel explicit coupling between the sectional aerosol model HAM-SALSA and the chemistry model MOZ was established to form isoprene derived secondary organic aerosol (iSOA). Isoprene oxidation in the chemistry model MOZ is described by a semi-explicit scheme consisting of 147 reactions, embedded in a detailed atmospheric chemical mechanism with a total of 779 reactions. Semi-volatile and low volatile compounds produced during isoprene photooxidation are identified and explicitly partitioned by HAM-SALSA. A group contribution method was used to estimate their evaporation enthalpies and corresponding saturation vapor pressures, which are used by HAM-SALSA to calculate the saturation concentration of each iSOA precursor. With this method, every single precursor is tracked in terms of condensation and evaporation in each aerosol size bin. This approach lead to the identification of dihydroxy dihydroperoxide (ISOP(OOH)2) as a main contributor to iSOA formation. Further, reactive uptake of isoprene epoxidiols (IEPOX) and isoprene derived glyoxal were included as iSOA sources. The parameterization of IEPOX reactive uptake includes a dependency on aerosol pH value. This model framework connecting semi-explicit isoprene oxidation with explicit treatment of aerosol tracers leads to a global, annual, average isoprene SOA yield of 15 % relative to the primary oxidation of isoprene by OH, $NO_3$, and ozone. With 445.1 Tg (392.1 TgC) isoprene emitted, an iSOA source of 138.5 Tg (56.7 TgC) is simulated. The major part of iSOA in ECHAM-HAMMOZ is produced by IEPOX 42.4 Tg (21.0 TgC) and ISOP(OOH)2 78.0 Tg (27.9 TgC). The main sink process is particle wet deposition which removes 133.6 (54.7 TgC). The average iSOA burden reaches 1.4 Tg (0.6 TgC) in the year 2012.

## 1 Introduction

Atmospheric particles play an important role in the earth system, especially in the interactions between climate (IPCC, 2013) and human health (Fröhlich-Nowoisky et al., 2016; Lakey et al., 2016). Aerosols interact with atmospheric radiation directly

via absorption and scattering, and indirectly via cloud formation. These interactions depend on the particles' microphysical properties, their chemical composition and phase state (Ghan and Schwartz, 2007; Shiraiwa et al., 2017). In the current political debates about air quality and climate change, understanding atmospheric particles is one of the most challenging problems and led to increased research in this field over the last two decades (Fuzzi et al., 2015). Especially organic aerosols are not well understood and subject to ongoing research (Pandis et al., 1992; Kanakidou et al., 2005; Zhang et al., 2007; Fuzzi et al., 2015; Hodzic et al., 2016). Organic aerosol (OA) consists of two types of particles, often mixed and difficult to distinguish (Kavouras et al., 1999; Donahue et al., 2009). First, organic aerosol can be emitted directly into the atmosphere as primary organic aerosol (POA) (Kanakidou et al., 2005; Dentener et al., 2006). Second, organic aerosol mass is also formed from organic gases which are emitted as volatile organic compounds (VOC) and transformed into compounds capable of partitioning into the particle phase. This second type of organic aerosol is called secondary organic aerosol (SOA) (Pankow, 1994; Seinfeld and Pankow, 2003; Jimenez et al., 2009).

Both types of organic aerosols are challenging to model due to limited knowledge about emissions, composition, evolution and physicochemical properties (Lin et al., 2012). Concerning SOA, there are additional uncertainties concerning SOA precursors and the atmospheric chemistry leading to their formation (Heald et al., 2005). Up to now, global models have lacked an explicit treatment of SOA (Zhang et al., 2007) and use relatively simple parameterisations to form SOA, for example with the two-product model by Odum et al. (1996). Such parameterisations neglect explicit chemical transformation and assume fixed SOA yields based on laboratory studies (Tsigaridis and Kanakidou, 2003; ODonnell et al., 2011). Donahue et al. (2006) presented with their volatility basis set (VBS) another approach that allows to distinguish between various precursor VOCs, but still does not consider explicit chemical formation and molecular identity of the compounds. The VBS system was further developed to include aerosol aging based on observations of O:C ratio (Donahue et al., 2011). Lin et al. (2012) and Marais et al. (2016) made first steps into coupling explicit formation of SOA precursors with SOA formation, focusing on specific compounds.

Global models largely underestimate the amount of atmospheric organic aerosol (Volkamer et al., 2006; De Gouw and Jimenez, 2009; Tsigaridis et al., 2014). This underestimation might be related to the huge number of organic compounds in the atmosphere (Goldstein and Galbally, 2007) which cannot be identified individually by state-of-the-art measuring devices. For explicit modeling, it is necessary to characterize their chemical properties, structures, volatility, solubility and further reactions pathways in the particle phase. Donahue et al. (2009) argues that it is extremely difficult to accomplish dissecting this complexity in detail.

This study makes an attempt to explore the influence of a semi-explicit chemical mechanism, implementing a state-of-the-art isoprene oxidation mechanism, which is based on Taraborrelli et al. (2009, 2012); Nölscher et al. (2014); Lelieveld et al. (2016), on isoprene derived secondary organic aerosol (iSOA) formation. Recently, isoprene was identified to contribute to SOA. Literature iSOA yields vary between 1% and 30% relative to the total amount of isoprene oxidized by OH, $O_3$, and $NO_3$ (Surratt et al., 2010). Even with a yield as low as 1%, isoprene as a source of SOA has a huge impact, since global annual isoprene emissions are estimated to range between 500 and 750 Tg $a^{-1}$ (Guenther et al., 2006). Therefore, iSOA was investigated in field and laboratory experiments (Claeys et al., 2004; Surratt et al., 2006, 2007a, b). These studies could identify

isoprene derived compounds in the particle phase and identified possible formation pathways (Liggio et al., 2005a; Lin et al., 2013b; Berndt et al., 2016; D'Ambro et al., 2017a). First generation products of isoprene are too volatile to partition into the the aerosol phase (Kroll et al., 2006), however they contribute significantly to iSOA formation via heterogeneous and multiphase reactions.

Glyoxal and isoprene epoxide (IEPOX) were identified to undergo reactive uptake and subsequent aqueous-phase reactions (Liggio et al., 2005b; Paulot et al., 2009). Glyoxal uptake might be followed by oligomerization and organo-sulfate formation depending on aerosol pH value, which is considered to be an irreversible uptake (Liggio et al., 2005a, b). Therefore, glyoxal derived SOA was studied in different model configurations with reversible and irreversible uptake (Volkamer et al., 2007; Fu et al., 2008; Ervens and Volkamer, 2010; Washenfelder et al., 2011; Waxman et al., 2013; Li et al., 2013).

Experimental and ambient measurements found 2-methyltetrol in the particle phase, which is attributed to be formed by IEPOX (Claeys et al., 2004; Paulot et al., 2009; Surratt et al., 2006). Therefore, irreversible reactive uptake from IEPOX was proposed. Surratt et al. (2007b) studied the effect of the pH value on iSOA formation and found the organic carbon mass as function of aerosol pH. This was studied further leading to the reaction mechanism for 2-methyltetrol formation from IEPOX to be an acid catalyzed ring opening reaction (Eddingsaas et al., 2010; Lin et al., 2013a) and was used to create

process parametrisations (Pye et al., 2013; Riedel et al., 2015). Nevertheless, IEPOX uptake was mostly studied in experiments using sulfate aerosol seeds to explore IEPOX uptake dependence on aerosol pH, which leads to the question whether the reaction might be sulfate catalyzed instead (Surratt et al., 2007a; Xu et al., 2015). However, non-racemic mixtures of tetrols stereoisomers in the atmosphere point to a substantial biological origin (Nozière et al., 2011).

    After exploring the IEPOX-SOA formation pathway, experimental studies could also identify non-IEPOX-SOA formation

pathways via the highly oxidized, rather low volatile isoprene product dihydoxy dihyodroperoxide ($C5H12O6$, ISOP(OOH)2) (Riva et al., 2016; Liu et al., 2016; Berndt et al., 2016; D'Ambro et al., 2017a). This compound was identified under low $NO_x$, meaning $HO_2$ dominated conditions (Berndt et al., 2016) and neutral aerosol pH (Liu et al., 2016; D'Ambro et al., 2017a).

    In light of the available knowledge on iSOA formation, this study focuses on iSOA formation via reactive uptake and explicit partitioning of exclusively semi- and low volatile isoprene derived compounds. This paper is organized as follows:

Section 2 describes the model framework including the sub-models ECHAM6, HAM-SALSA and MOZ. This includes a detailed description of the selection procedure for iSOA precursors and the interplay between the gas-phase oxidation of these and the SALSA aerosol scheme. Furthermore, Section 2 describes the model setup and sensitivity runs performed. Section 3 shows the simulation results of the reference run including all iSOA formation pathways, e.g. global annual budget and mean surface concentrations. Furthermore, additional process understanding is gained by several sensitivity simulations,

assessing the uncertainties in the reactive uptake of IEPOX, the isoprene oxidation mechanism, the saturation concentration and the evaporation enthalpy. Section 4 discusses possible error sources according to used parametrisations and assumptions and Section 5 provides conclusions.

## 2 Method

### 2.1 Model description

For this study, the aerosol chemistry climate model ECHAM-HAMMOZ in its version ECHAM6.3-HAM2.3MOZ1.0 is used (https://redmine.hammoz.ethz.ch/projects/hammoz/wiki/Echam630-ham23-moz10), Schultz et al. (2017)). This model frame-

work consists of three coupled models. ECHAM6 is the sixth generation climate model which evolved from the European Center for Medium Range Weather Forecasts (ECMWF) developed in the Max Planck Institute for Meteorology (Stevens et al., 2013). In order to simulate the climate, ECHAM6 solves the prognostic equations for vorticity, divergence, surface pressure and temperature expressed as spherical harmonics with triangular truncation (Stier et al., 2005). All tracers are transported with a semi Lagrangian scheme on a Gaussian grid (Lin and Rood, 1996). Hybrid $\sigma$-pressure coordinates with a pressure range

from 1013 hPa to 0.01 hPa are used for vertical discretization. Aerosol tracers are simulated by the Hamburg Aerosol Model HAM with aerosol microphysics based on the Sectional Aerosol module for Large Scale Applications SALSA (Kokkola et al., 2008; Bergman et al., 2012; Kokkola et al., 2018). In addition, the chemistry model MOZ simulates atmospheric concentrations of trace gases interacting with aerosols and the climate system (Stein et al., 2012). A detailed description of the HAMMOZ model system is given in Schultz et al. (2017).

For this study SALSA is extended to partition organic trace gases simulated by MOZ between the gas and aerosol phases. Additionally, the isoprene oxidation scheme in the MOZ chemical mechanism was modified in order to model secondary organic aerosol formation. Details can be found in Sections 2.1.1 and 2.1.2.

    Aerosol and trace gas emissions are taken from the ACCMIP interpolated emission inventory (Lamarque et al., 2010). Interactive gas phase emissions of VOCs are simulated by MEGAN (Model of Emissions of Gases and Aerosols from Nature)

(Guenther et al., 2006). For details about the implementation of MEGAN v2.1 in ECHAM-HAMMOZ and its evaluation the reader is referred to Henrot et al. (2017).

    For all simulations the triangular truncation 63, leading to horizontal resolution of $1.875° \times 1.875°$ and 47 vertical layers is used. The lowest layer, corresponding to the surface layer, thickness is around 50 m.

### 2.1.1 Chemistry model MOZ

Atmospheric chemistry is simulated by MOZ solving the chemical equations using an implicit Euler backward solver and treating emissions, dry and wet deposition. The current MOZ version evolved from an extensive atmospheric chemical mechanism based on MOZART version 3.5 (Model for Ozone and Related chemical Tracers) (Stein et al., 2012), which merges the tropospheric version MOZART-4 (Emmons et al., 2010) with the stratospheric version MOZART-3 (Kinnison et al., 2007). The chemical mechanism was further developed including a detailed isoprene oxidation scheme based on Taraborrelli et al. (2009,

2012); Nölscher et al. (2014); Lelieveld et al. (2016) with revised peroxy radical chemistry (Schultz et al., 2017), leading to a model system resembling the CAM-chem model (Community Atmosphere Model with Chemistry) (Lamarque et al., 2010). The chemical mechanism version used here is called JAM3 (Jülich Atmospheric Mechanism version 3). It differs from JAM version 2, evaluated in Schultz et al. (2017), in self and cross reactions of isoprene products, added nitrates, initial reactions for

**Table 1.** Isoprene oxidation products in JAM3, physical characteristics and molecular structure expressed as SMILES code. Pure-liquid saturation vapor pressure at the reference temperature 298 K $p_0^*$, Henry's law coefficient H and evaporation enthalpy $\Delta H_{vap}$. $\Delta H_{vap}$ and $p_0^*$ are used in Clausius-Clapeyron equation for calculation of the effective saturation vapor pressure as a function of temperature in SALSA. Names of the compounds rely on the Master Chemical Mechanism (MCM 3.2), except for LISOPOOHOOH, which is not in MCM 3.2. Names starting with "L" indicate that this specie is lumped, SMILES codes of all isomers are shown, but just the ones marked with * are used.

| Compound | SMILES code | $p_0^*(298.15K)$ [Pa] | $\Delta H_{vap}$ $\left[\frac{kJ}{mol}\right]$ | H $\left[\frac{mol}{atm}\right]$ |
|---|---|---|---|---|
| LNISOOH | O=CC(O)C(C)(OO)CON(=O)=O* | $2.2 \cdot 10^{-4}$ | 122.7 | $2.1 \cdot 10^5$ |
| | CC(O)(CON(=O)=O)C(OO)C=O | $3.8 \cdot 10^{-4}$ | 120.0 | |
| LISOPOOHOOH | OC(C)(COO)C(CO)OO* | $3.8 \cdot 10^{-7}$ | 155.3 | $2.0 \cdot 10^{16}$ |
| | CC(CO)(C(COO)O)OO | $1.9 \cdot 10^{-7}$ | 158.9 | |
| LC578OOH | OCC(O)C(C)(OO)C=O* | $2.0 \cdot 10^{-4}$ | 123.2 | $3.0 \cdot 10^{11}$ |
| | O=CC(O)C(C)(CO)OO | $2.0 \cdot 10^{-4}$ | 123.2 | |
| C59OOH | OCC(=O)C(C)(CO)OO* | $1.0 \cdot 10^{-4}$ | 125.0 | $3.0 \cdot 10^{11}$ |

monoterpenes and sesquiterepenes and production of low volatile, highly oxidized molecules. The additional isoprene related reactions can be found in Table S1. Similar extensions of terpene oxidation are planned; the current study focuses on isoprene. In total 254 gas species are undergoing 779 chemical reactions including 146 photolysis, 16 stratospheric heterogeneous and 8 tropospheric heterogeneous reactions. Thus, the 147 reactions in the semi-explicit isoprene oxidation scheme constitute a substantial fraction of these reactions in JAM3.

In order to identify SOA precursors produced via isoprene oxidation, first, a molecular structure was assigned to each chemical species. Some species are not represented explicitly, but instead they represent groups of compounds with similar chemical properties (lumping). In these cases one structure was assigned to the entire group of isomers. These structures are expressed as SMILES codes in Table 1 and as chemical structures in Figure 1. Second, with those molecular structures, the saturation vapor pressure $p^*(T)$ of each organic compound in JAM3 was estimated using the group contribution method by Nannoolal et al. (2008) and the boiling point method by Nannoolal et al. (2004) in the framework of the online open source facility UManSysProp (Topping et al., 2016). Third, the group contribution method data was fitted to the Clausius-Clapeyron equation in order to determine the evaporation enthalpy $\Delta H_{vap}$ for each compound. Finally, those species with saturation vapor pressures $p_0^*$ at 298.15 K lower than 0.01 Pa were classified as sufficiently low volatile to take their contribution for SOA formation into account. This procedure identified four isoprene oxidation products contributing to iSOA formation via gas-to-particle partitioning in ECHAM-HAMMOZ. Table 1 gives the SMILES codes and resulting pure-liquid saturation vapor pressure at the reference temperature $p_0^*$ and the evaporation enthalpy $\Delta H_{vap}$ for all iSOA precursors. The uncertainties in structure assignment of lumped species and the sensitivity to $\Delta H_{vap}$ are explored in Sections 3.2.3 and 3.2.4.

Figure 1 shows the chemical pathways of isoprene oxidation and their products to form LIEPOX, LNISOOH, LISOPOOH-OOH, LC578OOH and C59OOH. For the whole chemical mechanism including IGLYOXAL formation, the reader is referred to the model description of HAMMOZ in Schultz et al. (2017).

Isoprene derived SOA precursor gases are formed in MOZ via several reaction steps. Their formation is based on two initial reaction pathways from the oxidation of isoprene by OH and $NO_3$ respectively. The $O_3$ initiated reaction pathways are included in MOZ, but the products are too volatile to contribute to SOA formation. The OH initiated pathway leads to three iSOA precursors called C59OOH, LC578OOH and LISOPOOHOOH in our mechanism. First, OH attacks isoprene $C_5H_8$ and forms three isoprene peroxy radical isomers (R1), where one of them is a lumped specie. For simplicity they are called here ISOPO2.

$$C_5H_8 + OH \rightarrow ISOPO2 \tag{R1}$$

The ISOPO2 isomers either decompose (R2, Supplement 1, Table S1), undergo self- and cross-reactions (R3, Supplement 1, Table S1) and react with ambient radicals leading to isoprene hydroperoxides (ISOPOOH) (R4) and isoprene nitrates (ISOPNO3) (R5).

$$
\begin{aligned}
ISOPO2 &\rightarrow HO_2 + \text{other products} & \text{(R2)} \\
ISOPO2 + ISOPO2 &\rightarrow LHC4ACCHO + HCOC5 + HO_2 + \text{other products} & \text{(R3)} \\
ISOPO2 + HO_2 &\rightarrow ISOPOOH & \text{(R4)} \\
ISOPO2 + NO &\rightarrow ISOPNO3 & \text{(R5)}
\end{aligned}
$$

From the reactions of ISOPOOH with OH a hydroperoxide peroxy radical is formed, a lumped specie called LISOPOOHO2 (R6),

$$ISOPOOH + OH \rightarrow \alpha \cdot LISOPOOHO2 + \beta \cdot LIEPOX + \beta \cdot OH \tag{R6}$$

which can be oxidized by $HO_2$ to LISOPOOHOOH (R7). The stoichiometric coefficients $\alpha$ and $\beta$ vary depending on the ISOPOOH isomer which is oxidized. These stoichiometric coefficients can be found in Supplement 1 Table S2.

$$LISOPOOHO2 + HO_2 \rightarrow LISOPOOHOOH \tag{R7}$$

Not included in the JAM3 reference case is the 1,5 H-shift of LISOPOOHO2 that yields compounds with a higher volatility than LISOPOOHOOH (D'Ambro et al., 2017b), so the chemical yield of LISOPOOHOOH is expected to be an upper limit. D'Ambro et al. (2017b) estimated the rate of the 1,5 H-shift of LISOPOOHO2 to be higher than $0.1\,s^{-1}$. For this reason, the suggested product, an epoxide, might be more prevalent than LISOPOOHOOH, but still lead to a substantial

amount of iSOA via a similar heterogeneous reactive uptake like for IEPOX. The importance of LISOPOOHOOH and LISOPOOHO2 isomerization is discussed in Section 3.2.5, where the impact of 1,5 H-shift of LISOPOOHO2 is included in JAM3 and tested in two sensitivity simulations. Reactions (R1 – R7) show that LISOPOOHOOH production depends on ambient radical concentrations, thus it varies in space and time. On a global, annual, average for 2012, the chemical mass yield of LISOPOOHOOH is 9 %. This means that 9 % of the total carbon mass emitted in 2012 as isoprene end up as LISOPOOHOOH. LISOPOOHOOH can either react back to LISOPOOHO2, be photolysed or oxidized by OH to form LC578OOH (R8).

$$LISOPOOHOOH + OH \rightarrow LC578OOH + OH \tag{R8}$$

LC578OOH is a lumped species representing two MCM species C57OOH and C58OOH. LC578OOH is more volatile than LISPOOHOOH and can be formed via another pathway, as well.

$$
\begin{aligned}
ISOPO2 + NO &\rightarrow LHC4ACCHO + NO2 &\text{(R9)}\\
ISOPO2 + NO_3 &\rightarrow LHC4ACCHO + NO2 &\text{(R10)}\\
ISOPNO3 + h\nu &\rightarrow LHC4ACCHO + NO2 + HO_2 &\text{(R11)}\\
ISOPOOH + h\nu &\rightarrow LHC4ACCHO + OH + HO_2 &\text{(R12)}\\
LHC4ACCHO + OH &\rightarrow LC578O2 &\text{(R13)}\\
LC578O2 + HO_2 &\rightarrow LC578OOH &\text{(R14)}
\end{aligned}
$$

Reactions (R9 – R14) show LC578OOH formation via LHC4ACCHO degradation. LHC4ACCHO is a lumped species representing the MCM species HC4ACHO and HC4CCHO. Finally, LHC4ACCHO is oxidized by OH (R13) and forms LC578O2, which reacts with $HO_2$ to LC578OOH (R14). LC578OOH either reacts with OH back to LC578O2 or is photolysed. LC578O2 can undergo an 1,4 H-shift and recycle OH like for one of the RO2 from methacrolein (Crounse et al., 2011). On a global, annual, average for 2012, just 1 % of the oxidation of total isoprene carbon mass leads to LC578OOH.

The third compound formed from the OH initiated oxidation of isoprene is C59OOH. Starting from ISOPO2, there are two possible oxidation ways for C59OOH formation, one with nitrates as intermediates and a second one where nitrogen oxide is not required. The nitrate pathway starts with formation of ISOPNO3 from ISOPO2 (R5) and continues with OH reaction to form isoprene nitrate peroxy radicals ISOPNO3O2 (R15), which is again a lumped specie.

$$
\begin{aligned}
ISOPNO3 + OH &\rightarrow ISOPNO3O2 &\text{(R15)}\\
ISOPNO3O2 + CH3O2 &\rightarrow ISOPNO3OOH &\text{(R16)}\\
ISOPNO3OOH + OH &\rightarrow C59OOH &\text{(R17)}
\end{aligned}
$$

Via formation of a nitrate hydoxyperoxy radical, finally C59OOH is formed (R17). This pathway requires the availability of NO for the initial step in (R5). For the second pathway, no NO is needed.

$$HCOC5 + OH \quad \rightarrow \quad C59O2 \tag{R18}$$

$$C59O2 + HO_2 \quad \rightarrow \quad C59OOH \tag{R19}$$

Self-reactions of ISOPO2 (R3) lead to formation of HCOC5, which is then converted via OH to C59O2 (R18). $HO_2$ oxididses C59O2 to C59OOH (R19). C59OOH can also react back to C59O2 or be lost via photolysis. The overall, annual, average, mass yield from isoprene to C59OOH is 2 %.

The fourth iSOA precursor is an isoprene derived nitrate LNISOOH, which requires both, a $NO_x$ dominated and a $HO_2$ dominated environment, because only the first two oxidation steps use nitrate, then OH and $HO_2$ are required. First, isoprene

reacts with the $NO_3$ radical and forms a nitrate peroxy radical NIOSPO2 (R20), which oxidizes NO and forms NC4CHO (R21). NC4CHO in contrast, has to react with OH to form LNISO3 (R22), which then reacts with $HO_2$ and forms LINSOOH (R23).

$$C_5H_8 + NO_3 \quad \rightarrow \quad NISOPO2 \tag{R20}$$

$$NISOPO2 + NO \quad \rightarrow \quad NC4CHO \tag{R21}$$

$$NC4CHO + OH \quad \rightarrow \quad LNISO3 \tag{R22}$$

$$LNISO3 \quad \rightarrow \quad LNISOOH \tag{R23}$$

LNISOOH can be photolysed or react back to LNISO3. The fact that LINISOOH formation requires an environment where first NO dominates chemistry and then $HO_2$, limits its formation in the atmosphere. It is formed in in small amounts, therefore on an annual mean, oxidation of isoprene in 2012 yields only 0.1 % LNISOOH.

In Figure 1, a simplified overview over the described chemical reactions can be found. MOZ calculates branching ratios according to the ambient conditions, gas-phase yields shown in Figure 1 result from a global perspective. These gas-phase yields are resulting annual averages for 2012 and not fixed yields. Accordingly, the particle phase yields result from volatility or reactive uptake parametrization from the corresponding iSOA precursors. These yields are not fixed either, but are calculated from the global, annual average for the year 2012.

To cover multiphase chemical iSOA formation, heterogeneous reactions on aerosols of IEPOX and isoprene derived glyoxal were included. Nevertheless, ECHAM-HAMMOZ does not include in-particle or in-cloud aqueous phase chemistry, therefore no assumptions of in-particle products are made. Furthermore, no SOA formation via cloud droplets is included in ECHAM-HAMMOZ due to constraints in the aerosol cloud interaction formulation. Therefore, reactive uptake is parameterized as pseudo first order loss using aerosol surface area density given by HAM, according to Schwartz (1986) and described in detail

in Stadtler et al. (2017).

In MOZ, three IEPOX isomers are lumped together (LIEPOX) and a compound IGLYOXAL was introduced to be able to differentiate between isoprene derived glyoxal and glyoxal from other sources. Isoprene glyoxal formation pathways are numerous and no changes were made to the mechanism with respect to IGLYOXAL formation. Since these reactions are

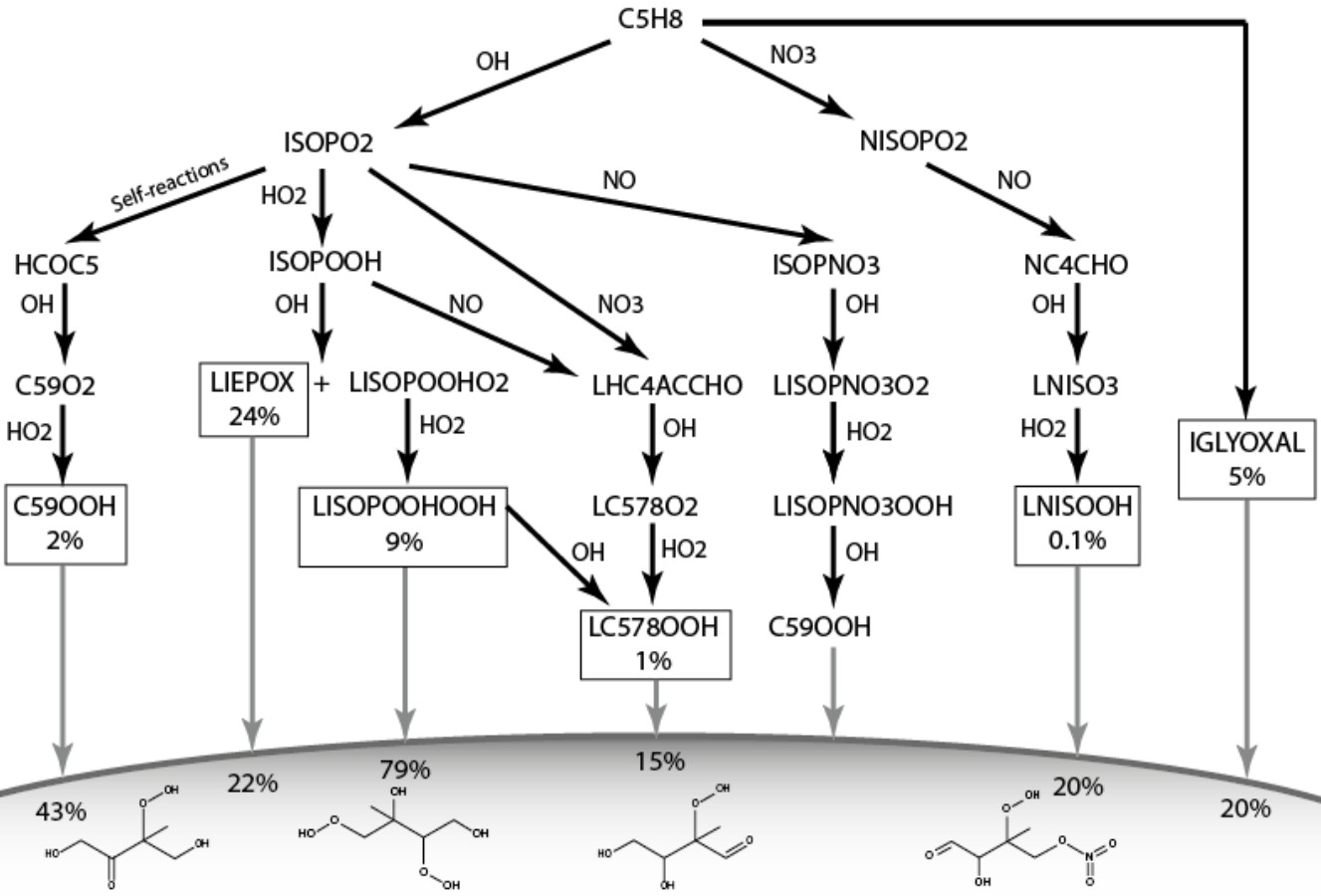

**Figure 1.** Simplified overview over chemical pathways leading to sufficiently low volatile isoprene derived compounds able to partition into aerosol phase. Note that ISOPO2 here is used for simplicity, JAM3 includes three different ISOPO2 (LISOPACO2, ISOPBO2, ISOPDO2), same applies for ISOPOOH. The percentages in the boxes indicate average mass yields, thus the annual, mean reaction turnover of isoprene leading to these products. For IGLYOXAL there are too many formation pathways and are therefore not shown. The solid horizontal curve represents the boundary to the particle phase. Percentages found under the corresponding arrow express the the annual, mean, individual, net iSOA yield of the compound. Except for LIEPOX and IGLYOXAL, structures are relevant to estimate the saturation vapor pressure and evaporation enthalpy and are therefore shown here. For the detailed mechanism, the reader is referred to Schultz et al. (2017).

included also in JAM2, see Schultz et al. (2017). LIEPOX is formed along the pathway described for LISPOOHOOH in reaction (R6).

Glyoxal is observed to produce a variety of compounds, like oligomers or organosulfates, in the aqueous aerosol phase and glyoxal is capable of being released back into the gas phase (Volkamer et al., 2007; Ervens and Volkamer, 2010; Washenfelder et al., 2011; Li et al., 2013). The simplification assuming irreversible uptake might thus overestimate its impact on iSOA. Following previous model studies (Fu et al., 2008; Lin et al., 2012) a reaction probability of $\gamma_{\mathrm{glyoxal}} = 2.9 \cdot 10^{-3}$ (Liggio et al., 2005b) is used.

For IEPOX the irreversibility is a less critical assumption, because IEPOX forms 2-methyltetrol and organosulfates in the aqueous aerosol phase, which stay in the aerosol phase (Claeys et al., 2004; Eddingsaas et al., 2010; Lal et al., 2012; McNeill et al., 2012; Woo and McNeill, 2015). However, ECHAM-HAMMOZ does not include explicit treatment of aqueous phase reactions. The reaction probability of IEPOX varies with pH value (Lin et al., 2013a; Pye et al., 2013; Gaston et al., 2014) which cannot be captured by ECHAM-HAMMOZ due to the lack of ammonium and nitrate in the aerosol phase, thus the possibility to capture aerosol pH. For these reasons, the reaction probability of IEPOX $\gamma_{\mathrm{IEPOX}} = 1 \cdot 10^{-3}$ (Gaston et al., 2014) was chosen, close to the value used by (Pye et al., 2013). To explore the impact of the pH dependence, sensitivity runs with different $\gamma_{\mathrm{IEPOX}}$ are analyzed. Additionally, no assumptions of in-particle products are made, in ECHAM-HAMMOZ IEPOX is simply taken up into the aerosol phase without further transformation.

### 2.1.2 HAM-SALSA

The Hamburg Aerosol Model (HAM) handles the evolution of atmospheric particles and includes emissions, removal, microphysics and radiative effects. Moreover, the current configuration uses the Sectional Aerosol module for Large Scale Applications (SALSA) for calculation of aerosol microphysics (Kokkola et al., 2008, 2018). In SALSA the aerosol size distribution is divided into aerosol size sections (size bins). Furthermore, these size bins are grouped into sub-ranges, which allows the model to limit the computation of the aerosol microphysical processes to include only the aerosol sizes that are relevant. Microphysical processes simulated by SALSA cover nucleation, condensation, coagulation, cloud activation, sulfate production and hydration (Bergman et al., 2012). The aerosol composition is described using five different aerosol compounds, sulphate, black carbon, dust, sea salt and organic carbon. Furthermore, SALSA treats secondary organic aerosol formation via the volatility basis set (Kühn et al., in preparation). In the model setup described there, SALSA uses a strongly simplified description for VOC oxidation (pseudo chemistry) to obtain SOA precursors. Here, the model system was extended and coupled to MOZ, which explicitly calculates SOA precursors as described in Section 2.1.1. The standard SALSA-VBS system is not used here. Instead for each SOA-forming compound the gas-to-particle partitioning is treated explicitly and its concentration is tracked in both the gas and the aerosol phases separately. This study exclusively uses isoprene-derived precursors to form iSOA, other oxygenated compounds capable of partitioning derived from terpenes or aromatics are neglected.

### 2.1.3 Coupling of HAM-SALSA and MOZ

HAM-SALSA and MOZ interact through several processes, oxidation fields calculated by MOZ are passed to HAM-SALSA for aerosol oxidation, MOZ produces H2SO4 which is then converted by HAM-SALSA to sulphate aerosol and HAM-SALSA provides the aerosol surface area density for heterogeneous chemistry. Above all, HAM-SALSA takes the information of iSOA precursor gas phase concentrations and their physical properties to calculate the saturation concentration coefficient ($C^*$) using Clausius-Clapeyron equation (1) (Farina et al., 2010).

$$C_i^* = C_i^*(T_0)\frac{T_0}{T} \exp\left[\frac{\Delta\text{H}_{vap}}{R}\left(\frac{1}{T_0} - \frac{1}{T}\right)\right] \tag{1}$$

Here $T_0$ is the reference temperature of 298.15 K and $\Delta\text{H}_{vap}$ is the evaporation enthalpy given in Table 1 for the iSOA precursors identified in this study. $C^*$ is then used to calculate the explicit partitioning of the iSOA precursors to each aerosol section. This process is reversible and it is thus possible that the iSOA formed in one region is transported and evaporates in another region. Explicitly calculating the partitioning instead of prescribing yields in chemical production or SOA formation is a key difference to other models with fixed yields. Loss processes for SOA in HAM-SALSA include sedimentation, deposition and wash out in the aerosol phase.

### 2.2 Simulation set up and sensitivity runs

An overview of the performed simulations can be found in Table 2. The reference simulation RefBase, which includes a three-month spin up and spans the time from October 2011 until the end of December 2012, is evaluated for the entire year 2012, while usually sensitivity runs are limited to the northern hemispheric, isoprene emission intense, summer season of June, July and August.

Several sensitivity simulations were performed to explore model sensitivities and assess uncertainties. For comparison of the explicit ECHAM-HAMMOZ scheme to a state-of-the-art VBS scheme, ECHAM-HAM whith pseudo chemistry and VBS configuration (RefVBS), described in 2.1.2, was run. ΔH30 uses the same, much lower evaporation enthalpy of $\Delta H_{vap} = 30\,\frac{\text{kJ}}{\text{mol}}$ for all partitioning species following Farina et al. (2010). The uncertainty in saturation vapor pressure estimation method was asses comparing Nannoolal et al. (2008) method to EVAPORATION (Compernolle et al., 2011) method.

Furthermore, pH value dependence of IEPOX is tested in $\gamma$IEX formulating an easy pH dependent parameterization based on laboratory measurements. Particle pH values cannot be obtained from ECHAM-HAMMOZ itself as the model does not include the calculation of particle phase thermodynamics. For this reason, aerosol pH was calculated offline using the AIM aerosol thermodynamics model (Clegg et al., 1998). The SALSA simulated annual mean mass of aerosol water and the mean mass of aerosol-phase inorganic compounds at the lowest model level were used an input for AIM. This required three additional assumptions: (1) all aerosol is in liquid form, (2) liquid water content is affected by all hygroscopic compounds, but only sulfate is assumed to affect the activity of the hygrogen ion (i.e. aerosol pH), (3) all sulfate is in form of ammonium bisulfate. This second assumption for sulfate has to be done, because particle phase ammonia is not modeled in the current configuration

of ECHAM-HAMMOZ. Using these inputs, AIM provided the concentration of the hydrogen ion ($H^+$) as an output. The resulting, global aerosol pH values thus vary strongly by region according to the RH and can be found in Figure S1 in the Supplementary 2.

As described in Section 2.1.1, JAM3 does not include the 1,5H-shift reaction of LISOPOOHO2. D'Ambro et al. (2017b)
describe the product resulting from 1,5H-shift reaction of LISOPOOHO2, a compound which is a highly oxidized epoxide. This compound is missing in ECHAM-HAMMOZ, thus the 1,5H-shift reaction was introduced as follows.

The structure of the compound described in D'Ambro et al. (2017b), relates to a compound which possibly undergoes the reactive uptake as IEPOX, but at the same time looks semi-volatile, like LC578OOH. For this reason, two simulations for the time period June, July and August 2012 were performed. One including reaction (R24).

$$LISOPOOHO2 \rightarrow LIEPOX \tag{R24}$$

And a second one including reaction (R25) instead of (R24). Both reactions use the best estimate for the reaction coefficient of $0.3\,s^{-1}$ (D'Ambro et al., 2017b). No temperature dependence was included.

$$LISOPOOHO2 \rightarrow LC578OOH \tag{R25}$$

To explore in-particle loss and SOA photolysis, short test runs including these processes were performed (DECAY and
JPHOT). In-particle loss is formulated as simple LISOPOOHOOH decay with a half-life of $4\,h$ (D'Ambro et al., 2017a; Stadtler, 2018). SOA photolysis is formulated like described in detail in Hodzic et al. (2015), but using a weaker photolysis frequency of $0.004\% J_{NO_2}$. This lower SOA photolysis frequency was chosen to take the argumentation by Malecha and Nizkorodov (2016) into account that in-particle photolysis is weaker than gas phase photolysis due to stabilization of the molecules in the particle.

**3 Results**

**3.1 Reference run RefBase**

**3.1.1 Global distributions**

Figure 2 shows the annual mean surface concentrations for total iSOA and its precursors in the gas phase. The precursors are formed, except for LNISOOH, during daytime and build up quickly. Therefore, these are found very close to isoprene
source regions mostly in the tropics and southern hemisphere. Their highest values, up to $3\,\mu g\,m^{-3}$, are simulated over the Amazon, the east flank of the Andes, Central Africa, North Australia, Indonesia and Southeast Asia. In the annual mean also the northern hemispheric summer is visible, but peak values of over $2\,\mu g\,m^{-3}$ are only reached on Mexico's west coast and in the Southeastern US. In Europe and North Asia, where isoprene emissions are much lower, mean values up to $0.5\,\mu g\,m^{-3}$ of precursors are formed.

**Table 2.** Description of simulations performed.

| Simulation | Description | Simulation period |
|---|---|---|
| RefBase | Reference run with uniform reaction probabilities for IEPOX and isoprene glyoxal $\gamma_{IEPOX} = 1.0 \cdot 10^{-3}$, $\gamma_{IGYOXAL} = 2.9 \cdot 10^{-3}$ (see Section 2.1.1), Partitioning precursor $\Delta H_{vap}$ and $p_0^*(298.15K)$ given in Table 1. | Whole year 2012 |
| RefVBS | ECHAM-HAM simulation with VBS approach and pseudo chemistry (see Section 2.1.2). | June, July, August 2012 |
| $\Delta$H30 | Like RefBase, but with same $\Delta H_{vap} = 30 \frac{kJ}{mol}$ for all compounds. | June, July, August 2012 |
| EVA | Like RefBase, but with $\Delta H_{vap}$ and $p_0^*$ derived with EVAPORATION (Compernolle et al., 2011) instead of Nannoolal et al. (2008) method. | Whole year 2012 |
| $\gamma$pH | Like RefBase, but with $\gamma_{IEPOX} = f(pH)$. | June, July, August 2012 |
| HshiftIEP | Additional reaction in JAM3 (R24). | June, July, August 2012 |
| HshiftLC5 | Additional reaction in JAM3 (R25). | June, July, August 2012 |
| DECAY | LISOPOOHOOH in-particle decay. | June, July, August 2012 |
| JPHOT | SOA photolysis with $J_{SOA} = 0.004\% J_{NO_2}$. | June, July, August 2012 |

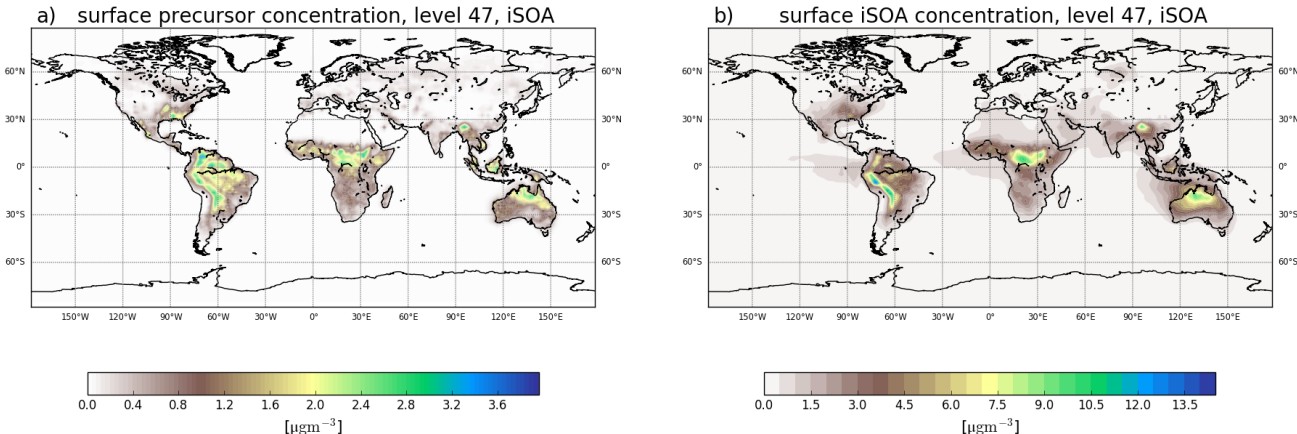

**Figure 2.** Global surface layer maps showing, the iSOA precursor gas phase concentration in plot a) and the aerosol phase iSOA concentration in plot b) as annual averages for 2012 in $\mu g \, m^{-3}$. The reader should note the different color scales, higher concentrations are reached in the aerosol phase, plot b).

These low precursor concentrations correspond to the very low iSOA concentrations over Europe and North Asia, compared to Southeastern US and Mexico's west coast, where up to $4.5\,\mu g\,m^{-3}$ iSOA is formed. The highest iSOA concentrations are found where high precursor concentrations meet pre-existing aerosol, like in Central Africa because of it high biomass burning emissions or Southeast Asia, where aerosol pollution is high. In the latter ECHAM-HAMMOZ simulates values of up to

$\quad$ $13\,\mu g\,m^{-3}$ iSOA. The Amazon is a region of very high isoprene emissions and therefore high iSOA precursor concentrations, nevertheless the local maximum in iSOA of $13.5\,\mu g\,m^{-3}$ can be seen on the east side of the Andes. This pattern is caused by pre-existing aerosol, which in ECHAM-HAMMOZ tends to accumulate on the east side of the Andes, and the still high iSOA precursor concentrations in the same region. Also in the northern part of Australia higher precursor loadings are found, leading there to iSOA ground-level concentrations of up to $9\,\mu g\,m^{-3}$.

$\quad$ It can also be seen, that iSOA has a longer lifetime than its gas phase precursors. Prevailing wind directions are recognizable, clearly showing transport of iSOA over the oceans, for example in the South American and African outflow regions. Also, iSOA is transported from Australia to the north. The average iSOA lifetime was calculated to be around 4 days, so long-range transport is limited, before iSOA is lost due to wet deposition (see Section 3.1.2).

$\quad$ Farina et al. (2010) included iSOA formation with fixed yields of isoprene transformation to the different VBS classes and

$\quad$ also showed its global annual surface distribution for $1979-1980$. Compared to Farina et al. (2010) ECHAM-HAMMOZ simulates nearly one order of magnitude higher maximum iSOA concentrations. This is explained by much higher reaction turnover from MOZ leading to higher amounts of iSOA precursors than produced by the low yields prescribed in Farina et al. (2010). The global patterns agree in high values over Southeastern US, South America, Central Africa and North Australia. In contrast, Farina et al. (2010) do not simulate the maximum over Southeast Asia. Hodzic et al. (2016) show biogenic SOA

$\quad$ for the lower 5 km on a global scale, again general patterns agree with the distribution in ECHAM-HAMMOZ. Nevertheless, Hodzic et al. (2016) simulated higher concentrations over Eurasia, which is not captured by ECHAM-HAMMOZ due to the lack in other biogenic VOC derived SOA.

$\quad$ Total biogenic SOA concentrations with iSOA surface concentrations of ECHAM-HAMMOZ compare well with their order of magnitude, which again underlines the higher yields resulting in ECHAM-HAMMOZ. High concentrations result from the

$\quad$ highly oxidized compounds produced by MOZ chemistry, especially due to LISOPOOHOOH which has a molar mass of $168.14\,g\,mol^{-1}$ that is very large. LISOPOOHOOH and LIEPOX contribute most to iSOA, followed by isoprene glyoxal. To further discuss this, iSOA composition concentrations for northern hemispheric summer (June, July, August) are shown in Figure 3.

$\quad$ Figure 3 a), c), e), g) show gas phase precursor concentrations and c), d), f), g) show particle phase concentrations. First,

$\quad$ LIEPOX a), b) and LISOPOOHOOH c), d) are shown, because they have greatest impact in the particle phase, followed by IGLYOXAL e),f). The other iSOA precursors are shown together because of their low concentrations in gas g) and particle phase h). On the right hand side corresponding mean values in the particle phase are displayed.

$\quad$ From the gas phase LIEPOX distribution a) it can be seen that MOZ simulates concentrations of around $0.5\,\mu g\,m^{-3}$ over isoprene rich areas. Peak values of $4.5\,\mu g\,m^{-3}$ LIEPOX are found over Southeastern US, north Venezuela and north of

$\quad$ Myanmar. Higher concentrations of LIEPOX are reached in the aerosol phase. For example, in South America gas phase

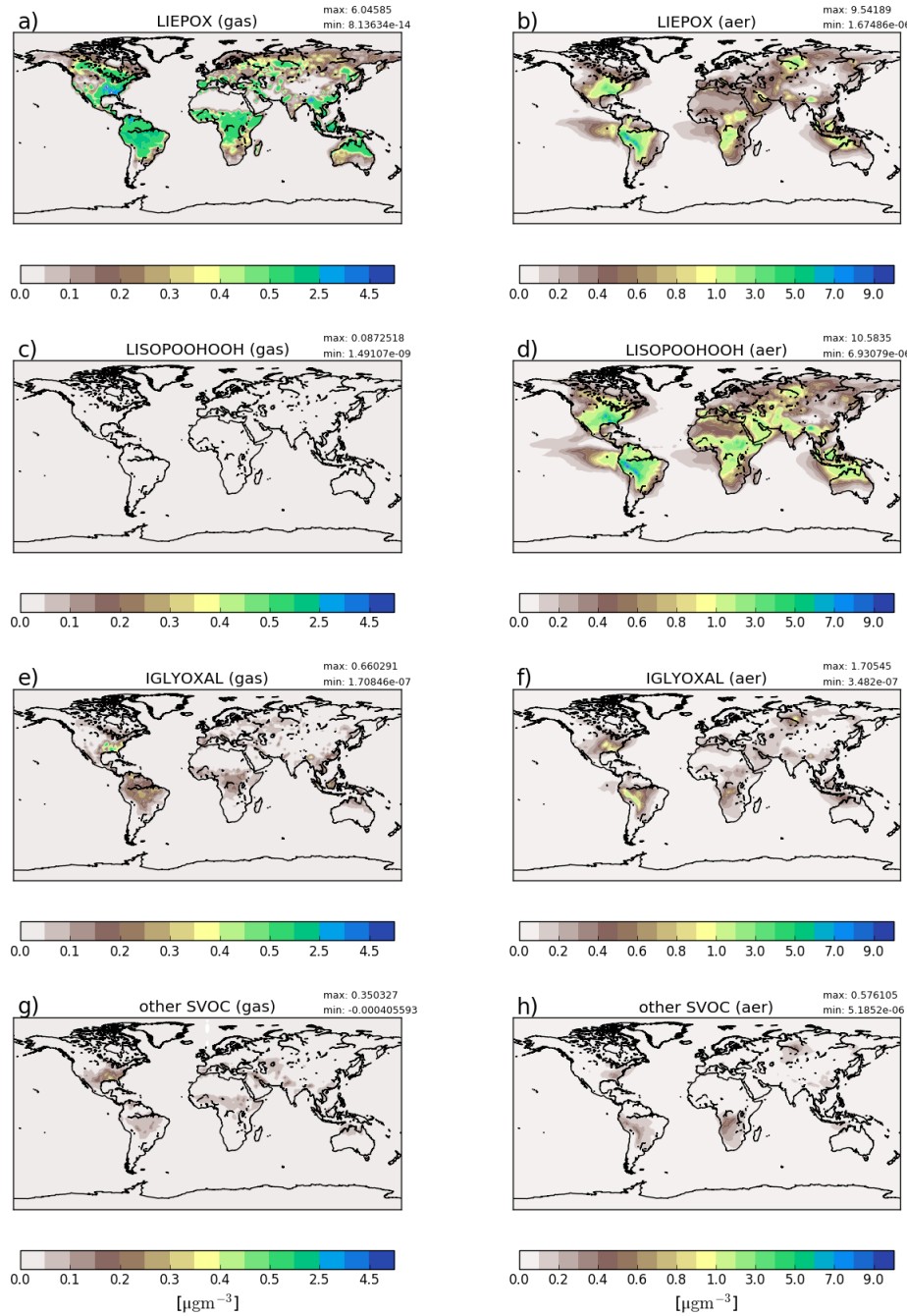

**Figure 3.** Reference run average surface distribution of precursor gases (a, c, e, g) and corresponding component concentration in the particle phase (b, d, f, h) in $\mu g\,m^{-3}$ for June, July and August 2012. Since concentrations of non-LISPOOHOOH-iSOA is below $1\,\mu g\,m^{-3}$, they are shown together in g) and h). Different scales are used for precursors and iSOA to capture the concentration ranges accordingly. Note, that the concentration scales are not linear and focus on low concentrations.

concentrations varies between 1.5 and 2.5 $\mu g\,m^{-3}$, but LIEPOX-SOA values over 7 $\mu g\,m^{-3}$ are reached on the eastern edge of the Andes. Especially, LIEPOX-SOA transport over the ocean and over Sahara can be seen. No assumption of in-particle products for LIEPOX was made, but usually 2-methyltetrols are present in $ng\,m^{-3}$ concentrations in the particle phase (Claeys et al., 2004; Kourtchev et al., 2005; Clements and Seinfeld, 2007). Lopez-Hilfiker et al. (2016) report that 80 % of IEPOX forms dimers instead of 2-methyltetrols, which would increase the concentrations of IEPOX derived SOA in the ambient measurements. Accounting additionally for these 80 %, the mass concentrations would reach around $10-100\,ng\,m^{-3}$, still an overestimation of simulated LIEPOX-SOA is indicated.

In contrast to LIEPOX, LISOPOOHOOH gas phase concentrations c) are very low and even with a scale focusing on low values, these cannot be captured on a scale fitting to the other compounds. For the gas phase LISOPOOHOOH globally lower values than $0.1\,\mu g\,m^{-3}$ are calculated. This is a consequence of iSOA formation. Figure 3 d) shows that LISOPOOHOOH-SOA appears in high concentrations between 1 and 8 $\mu g\,m^{-3}$ in the particle phase, because of its low volatility. Depending on the region, even more iSOA is formed by LISOPOOHOOH than LIEPOX, like over the Middle East. Indeed, the sum of LIEPOX and LISOPOOHOOH mass concentration comprise up to 90 % of the iSOA mass.

IGLYOXAL e), f) and the sum of the other partitioning iSOA precursors g),h) show similar global distributions and concentrations. Nevertheless, reactive uptake is more efficient producing more IGLYOXAL-SOA f) than from the other partitioning iSOA precursors h). IGLYOXAL shows similar maxima as LIEPOX over the American continent, north of Myanmar and over Siberia. Whereas the sum of other partitioning iSOA precursors shares areas of peak values with LISOPOOHOOH, pointing out the different iSOA formation processes. Similarly, up to 8 % of iSOA is formed by IGLOYXAL, the rest of 2 % mainly consist of C59OOH.

Particle concentrations seem high taking into account, that possible isoprene IVOC and more volatile SVOC were excluded. Hodzic et al. (2016) hypothesize that modeled SOA concentrations might compare better to observed SOA levels, if a faster turnover was simulated. This includes, a stronger SOA formation, but also a stronger removal. Currently, global models usually account for deposition loss, but ignore removal processes as fragmentation, aqueous phase reactions and in-particle photolysis. The next Section 3.1.2 compares iSOA production to SOA production of the AeroCom models and explains the high concentrations described here. Including more aerosol sinks following Hodzic et al. (2016) could reduce these concentrations even if iSOA production remains as it is currently simulated (see Section 3.2.5.

To summarize, Figure 3 shows that particle formation does not only depend on precursor concentrations, but also on available preexisting aerosol. Since all compounds are produced by isoprene the global distribution of the individual gases does not differ a lot. In contrast to the annual mean, the Northern Australian maximum does not appear that prominently during the northern hemispheric summer. Hence, the great impact in Northeastern US is clearly visible. For Europe, even during summer, iSOA seems to play a minor role compared to the equatorial regions due to prevalent vegetation (Steinbrecher et al., 2009).

### 3.1.2 Global iSOA budget

The global annual budget for isoprene derived secondary organic aerosol is shown in Figure 4. For the evaluated simulation period of 2012 a total of 392.1 TgC isoprene were emitted, which is a bit lower than the range of estimated isoprene emissions

**Table 3.** Total annual chemical production of individual iSOA precursors 2012 and corresponding amount of iSOA formed. In parenthesis the corresponding yields are given, for the gas phase how much of total isoprene was converted to the precursors and the yield of those precursors into iSOA for the global annual budget.

| Specie | Gas phase production in TgC (fraction of isoprene source) | Particle formation in TgC (individual yield in %) |
|--------|------------------------------------------------------------|---------------------------------------------------|
| LIEPOX | 94.0 (24 %) | 21.0 (22 %) |
| IGLYOXAL | 19.8 (5 %) | 3.6 (20 %) |
| LISOPOOHOOH | 35.1 (9 %) | 27.9 (79 %) |
| C59OOH | 6.5 (2 %) | 2.8 (43 %) |
| LC578OOH | 4.5 (1 %) | 0.3 (15 %) |
| LNISOOH | 0.5 (0.1 %) | 0.1 (20 %) |

$440 - 660$ TgC (Guenther et al., 2006; Henrot et al., 2017). The oxidation of isoprene leads to production of 160.4 TgC of the six iSOA precursors identified in this study. Comparing it to the initially emitted amount, 41 % of isoprene is chemically transformed into iSOA precursors. 24 % of isoprene end up as IEPOX, 9 % as LISOPOOHOOH, 5 % as IGLYOXAL, 2 % as C59OOH, 1 % as LC578OOH and 0.1 % as LNISOOH (see Table 3). For LIEPOX 94.0 TgC are produced, which agrees very

well with $95 \pm 45$ TgC estimated by Paulot et al. (2009). Of the total produced iSOA precursors, about a third (56.7 TgC) form iSOA. Half of iSOA is formed by reactive uptake, where IEPOX contributes 21.0 TgC and glyoxal 3.6 TgC, corresponding to a reactive uptake yield of 22 % (LIEPOX) and 20 % (IGLYOXAL), respectively. Since the reactive uptake is irreversible and the partitioning species are semi and low volatile compounds, evaporation is several orders of magnitude lower than condensation. This results in an annual overall isoprene SOA yield of 15 %, and in a global burden of 0.6 TgC. A isoprene SOA yield of

15 % lies in the range of 1 % to 30 % under different conditions observed by Surratt et al. (2010). Sinks of the precursor gases are chemical loss including photolysis, dry and wet deposition. The majority of precursors is destroyed chemically, the second most important sink is wet deposition. Aerosols can be lost via three processes in ECHAM-HAMMOZ, via sedimentation, dry and wet deposition. For iSOA sedimentation is 0.2 TgC and is for a clearer structure not included in Figure 4. The main loss of iSOA is wet deposition removing 54.7 TgC of the total of 56.7 TgC.

Table 4 shows the iSOA budget in Tg to be comparable with the mean values of the AeroCom (Aerosol Comparisons between Observations and Models) given in Tsigaridis et al. (2014). As can be seen from Table 4, the iSOA production of ECHAM-HAMMOZ in the reference simulation exceeds total SOA of the AeroCom models in the upper third quartile limit. Even if this comparison here seems to show a vast overestimation by ECHAM-HAMMOZ 56.7 TgC iSOA do not reach the lower end of the top down estimated source strength ranging from $140 - 910$ TgC a$^{-1}$ (Goldstein and Galbally, 2007; Hallquist

et al., 2009). Therefore, according to these studies, the AeroCom models generally produce too little SOA, while our new approach might lead to more realistic SOA concentrations. Using the range of $140 - 910$ TgC a$^{-1}$ for total SOA and our iSOA production of 56.7 TgC a$^{-1}$ would imply that isoprene contributes between 6% and 41 % to total SOA. This does not seem unrealistic. Dry deposition and wet deposition are higher than the AeroCom mean value because the iSOA burden is larger.

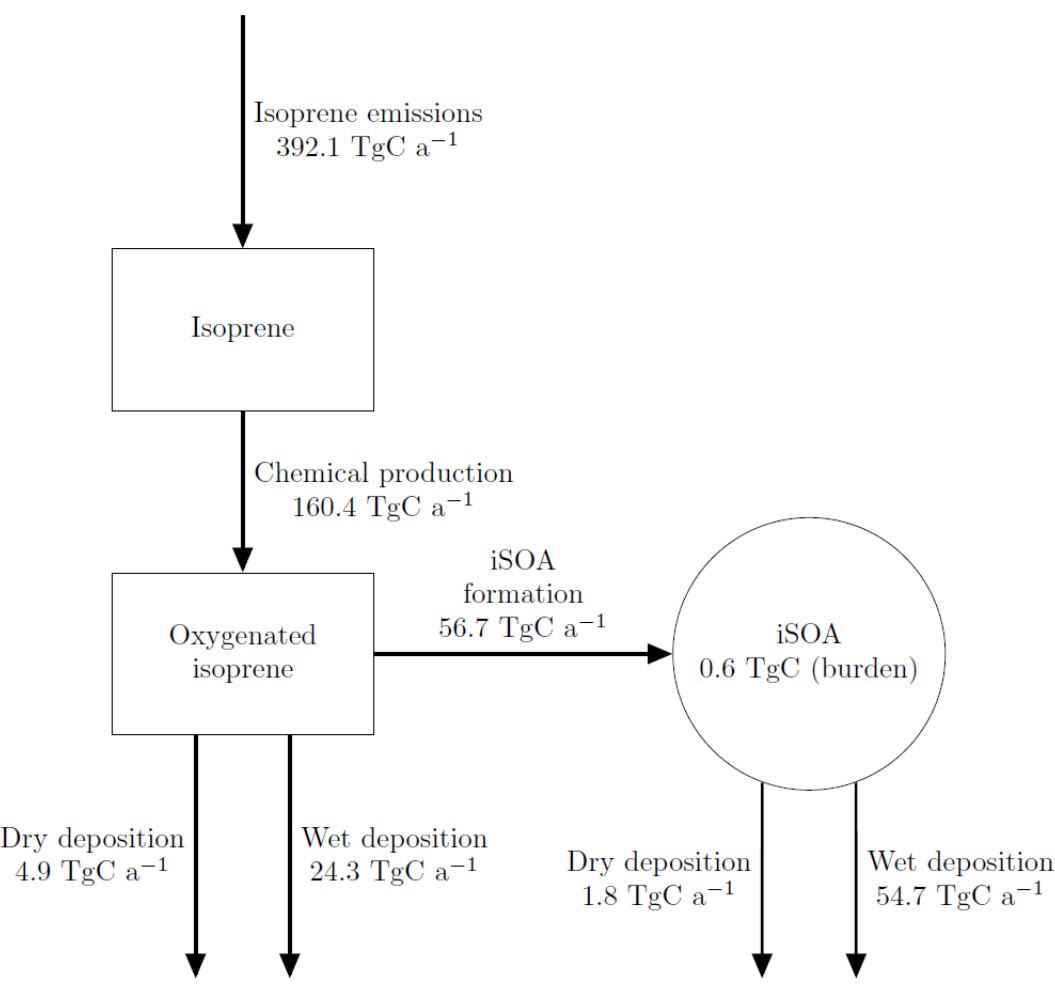

**Figure 4.** Global budgets for isoprene derived secondary organic aerosol and its precursors (sources/sinks in TgC a$^{-1}$ and burden in TgC) predicted by ECHAM-HAMMOZ reference simulation for 2012. For details about the individual compounds see Table 3.

Nevertheless, in ECHAM-HAMMOZ wet deposition is more than ten times higher than dry deposition, something that is not seen in the AeroCom models. First, this might point to a too low aerosol dry deposition in ECHAM-HAMMOZ. Second, high wet deposition might be caused by moisture and convection overestimation of ECHAM6 in the tropical regions where most of iSOA is formed. Finally, the iSOA burden in ECHAM-HAMMOZ is also higher than the mean of AeroCom models, while

5    iSOA lifetime of 3.7 days is in the lower end. The comparably short lifetime of 3.7 days is mainly caused by the quick wet deposition loss of iSOA. In ECHAM-HAMMOZ, iSOA is produced in tropical regions with high relative humidity and active convection, which trigger the wet deposition loss of particle phase iSOA near to the source regions.

As stated in Hodzic et al. (2016), global models are missing aerosol sinks like in-particle fragmentation and particle photolysis and should therefore overestimate SOA formation. On the contrary, global models tend to underestimate SOA formation.

**Table 4.** Comparison of the ECHAM-HAMMOZ iSOA budget to total SOA budget terms from AeroCom models (annual OA budget like in Figure 1 Tsigaridis et al. (2014), personal communication).

|                | ECHAM-HAMMOZ | AeroCom mean | AeroCom range |
|----------------|--------------|--------------|---------------|
| Sources        | $138.5\,\mathrm{Tg\,a^{-1}}$ | $36.3\,\mathrm{Tg\,a^{-1}}$ | $12.7\text{–}120.8\,\mathrm{Tg\,a^{-1}}$ |
| Dry deposition | $4.4\,\mathrm{Tg\,a^{-1}}$ | $5.7\,\mathrm{Tg\,a^{-1}}$ | $1.4\text{–}14.5\,\mathrm{Tg\,a^{-1}}$ |
| Wet deposition | $133.6\,\mathrm{Tg\,a^{-1}}$ | $47.9\,\mathrm{Tg\,a^{-1}}$ | $12.4\text{–}113.1\,\mathrm{Tg\,a^{-1}}$ |
| Burden         | $1.4\,\mathrm{Tg}$ | $1.0\,\mathrm{Tg}$ | $0.3\text{–}2.3\,\mathrm{Tg}$ |
| Lifetime       | 3.7 days | 8.2 days | 2.4–14.8 days |

The comparison of ECHAM-HAMMOZ iSOA to total SOA of other models shows that the criticized underestimation is more than resolved, since no SOA from aromatics or terpenes is considered in this study. Including semi-explicit chemistry and explicit partitioning leads in ECHAM-HAMMOZ to a high isoprene SOA yield, which motivated several sensitivity runs.

## 3.2 Sensitivity runs

### 3.2.1 Comparison to pseudo chemistry SOA

In order to compare the differences in the atmospheric iSOA loads when using the novel coupling of SALSA and MOZ with detailed iSOA chemistry (see Section 2.1.3) or when using a more parameterised VBS approach, an ECHAM-HAM simulation which applies VBS (RefVBS) for iSOA formation was run. All input parameters in RefBase and RefVBS are the same, especially the climate model ECHAM6 is exactly the same version. However, the major difference is the calculation of atmospheric chemistry. ECHAM-HAM, when it is not coupled to MOZ, uses parameterisations for sulfate aerosol formation and reads in offline fields for oxidant concentrations, while HAM in ECHAM-HAMMOZ obtains chemical information from MOZ. Furthermore, the SOA precursor formation approaches differ. As explained in Section 2.1.2 ECHAM-HAM with SALSA uses fixed yields to form SOA precursors from the oxidation reactions. Thus, differences in iSOA precursors and iSOA concentrations are caused by the differences in the level of sophistication in solving the atmospheric chemistry. Furthermore, the volatilities of the SOA precursors in the two models differ, which will be discussed in more detail below.

To compare the semi-explicit chemistry and explicit compound-wise partitioning to the pseudo chemistry and VBS system, an ECHAM-HAM run was performed just including isoprene emissions to form only iSOA in both models. From these isoprene emissions ECHAM-HAM produces gas phase compounds of the VBS classes VBS0, VBS1. In these simulations the yield for non-volatile SOA precursors is set to zero and thus no VBSnonvol is formed from isoprene. VBS0 and VBS1 refer to compounds with $\log 10(C^*) = 0$ and $\log 10(C*) = 1$ ($C^*$ in $\mathrm{\mu g\,m^3}$) respectively. VBS0 and VBS1 are classes containing SVOC, comparable to ECHAM-HAMMOZ C59OOH, LC578OOH and LNISOOH. Additionally, ECHAM-HAMMOZ includes the compound LISOPOOHOOH, which would be attributed to the class VBSnonvol, but as mentioned above, such a low volatile compound is not produced from isoprene in the ECHAM-HAM with VBS. Further, ECHAM-HAM does not include IEPOX and glyoxal SOA, thus here these two compounds are not included in this comparison, although they con-

tribute to iSOA in RefBase. The differences between the chemical production of the SOA precursors in ECHAM-HAMMOZ and ECHAM-HAM lead to differences in the amount of compounds of different volatilities. ECHAM-HAMMOZ chemistry yields very high amounts of LISOPOOHOOH and less of other SVOCs. In contrast, the pseudo-chemistry in ECHAM-HAM with VBS only leads to the formation of SVOCs from isoprene chemistry, lacking the compounds of lowest volatility. Total iSOA formed by partitioning including SVOC from ECHAM-HAM RefVBS is compared to iSOA from SVOC and LVOC in ECHAM-HAMMOZ reference run RefBase.

The formed precursors in the gas phase from RefVBS compared to the precursors from RefBase are shown in Figure 5. From the higher gas phase concentrations (Figure 5 b)), it can be seen that the VBS system only includes semi volatile compounds. The emission pattern of MEGAN is clearly visible in both model results, but in RefBase some isoprene emitting areas are hard to distinguish, because there the concentrations are very low.

Nevertheless, the low gas phase concentrations in RefBase do not mean, that less iSOA precursors were formed. On the contrary as can be seen in Figure 5, iSOA from SVOC and LVOC in RefBase is overall higher and horizontally transported further than iSOA in RefVBS. Local maxima match between both models, the higher values in Southeastern US and in the Amazon are captured by both models. However, in Southeastern US RefBase simulates values around $6\,\mu g\,m^{-3}$ over a broader area than RefVBS reaching $3.5\,\mu g\,m^{-3}$ in two more local maxima. Similarly, over the Amazon and north of the Andes RefBase simulates up to $9\,\mu g\,m^{-3}$ while RefVBS reaches $3\,\mu g\,m^{-3}$. Both simulations also agree on a local maximum in Central Africa and over North Australia and Indonesia. Again, peak concentrations differ, here by a factor of around 2.

RefVBS includes more SVOC than RefBase, leading to an equilibrium in RefVBS between gas phase and aerosol phase iSOA, which favors higher gas phase concentrations than seen in RefBase. This results from different chemical precursor formation, with the semi-explicit MOZ forming on average lower-volatile SOA precursors, which favor partitioning to the particle phase. LISOPOOHOOH formation is not taken into account in the ECHAM-HAM pseudo-chemistry formulation, which only forms SVOC, and explains the comparably low iSOA yields. Additionally, LIEPOX- and IGLYOXAL-SOA, which are not shown in Figure 5, but are included in RefBase, lead to increased SOA mass in RefBase compared to RefVBS. Increased aerosol mass increases the SOA yield. This could be another reason why more organic mass partitions in the particle phase in RefBase than in RefVBS.

To summarize, given the same isoprene emissions, the ECHAM-HAM pseudo-chemistry produces less iSOA precursors with an average higher volatility compared to the semi-explicit chemistry in MOZ, which results in a higher overall iSOA yield in ECHAM-HAMMOZ. Moreover, ECHAM-HAM does not include IEPOX- and glyoxal-SOA, which may positively affect the gas-to-particle partitioning of the volatile SOA species. This explains the higher precursor concentrations and the lower iSOA concentrations in ECHAM-HAM compared to ECHAM-HAMMOZ. In order to get similar iSOA and precursor concentrations, ECHAM-HAM pseudo chemistry could be adjusted accordingly.

### 3.2.2 IEPOX sensitivity to aerosol pH

As discussed in Section 2.1.1, several laboratory and field studies suggested a pH value influence on the reactive uptake of IEPOX. ECHAM-HAMMOZ does not include ammonium and nitrate aerosol, therefore no aerosol pH value can be obtained

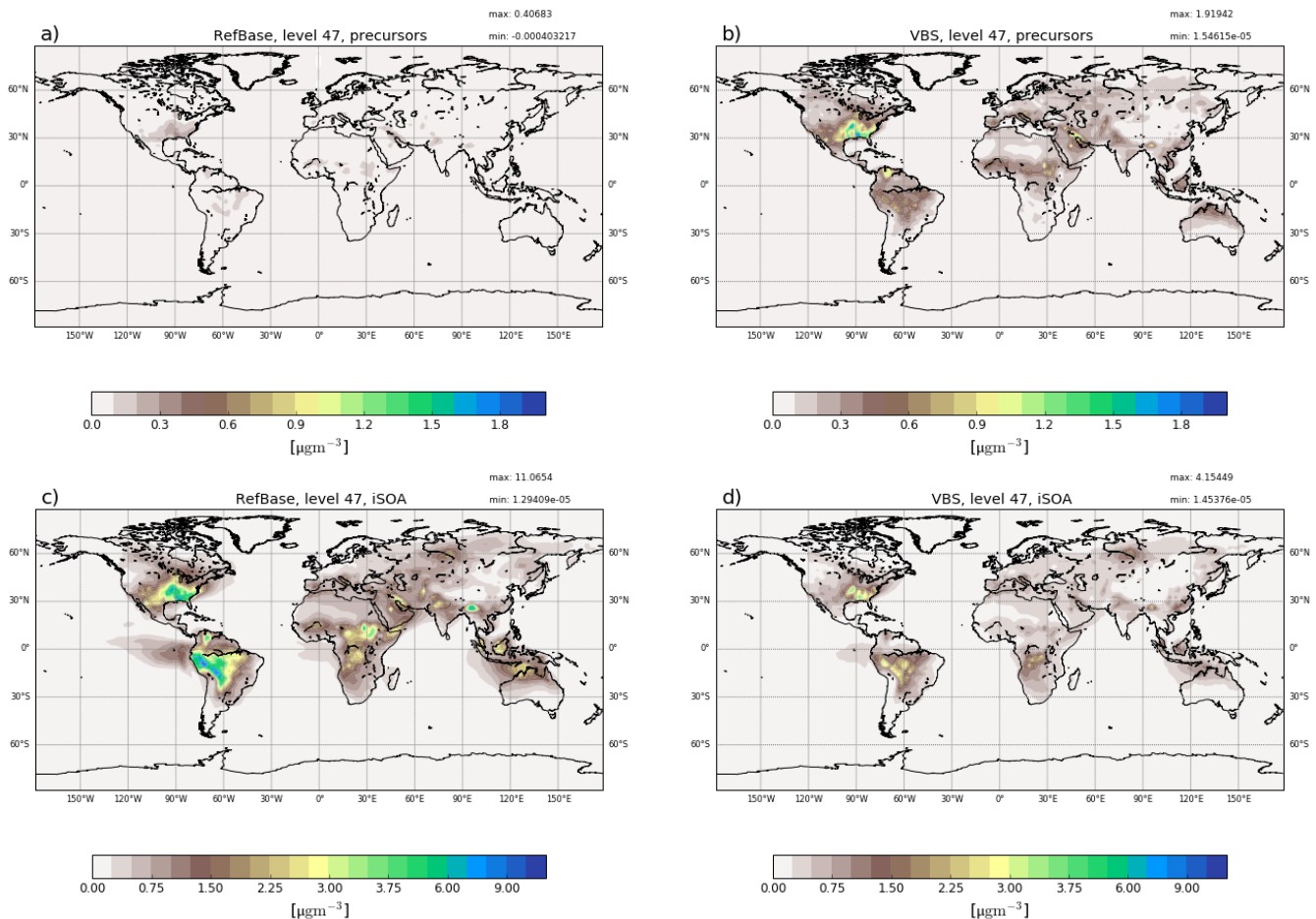

**Figure 5.** Surface, average concentrations of gas phase iSOA precursors (a and b) and aerosol phase iSOA (c and d) for June, July and August 2012. Plots a) and c) show concentrations simulated by the reference run RefBase and plots b) and d) the concentrations calculated by RefVBS using pseudo chemistry and the VBS system. For RefBase the precursors consist of the four iSOA precursors (LNISOOH, LC578OOH, C59OOH, LISOPOOHOOH) described above, for RefVBS concentrations the sum of gas phase VBS classes VBSnonvol, VBS0, VBS1 is shown.

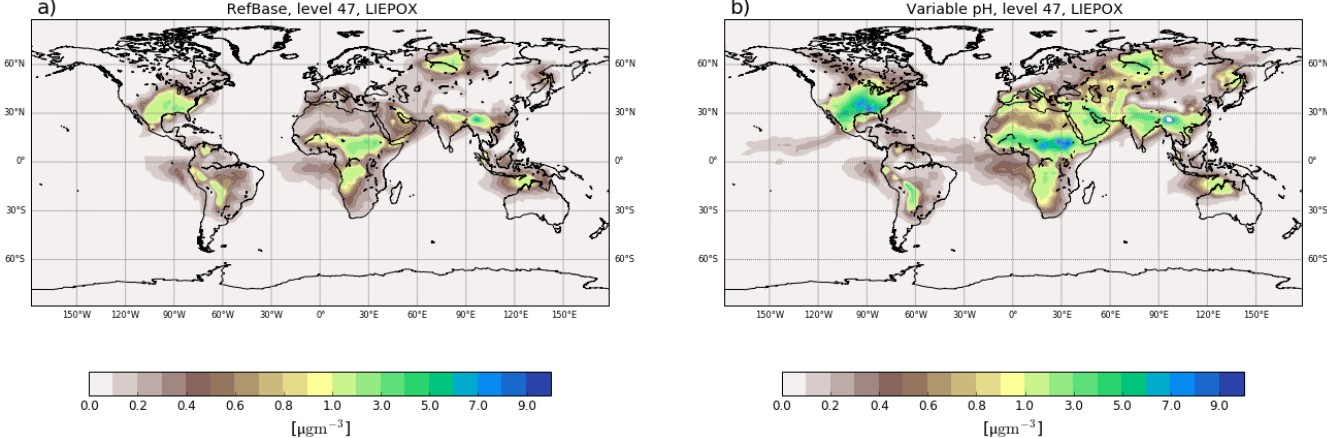

**Figure 6.** Surface, average aerosol concentrations in $\mu g\,m^{-3}$ for LIEPOX derived iSOA with uniform pH value used in the reference run (a) and with variable pH value calculated with AIM aerosol thermodynamics model (see Section 2.2) in the sensitivity run $\gamma$pH (b) for the time period of June, July and August 2012.

by the model system. As described in Section 2.2 a simulation with AIM aerosol thermodynamics model was performed to obtain the global aerosol pH distribution consistent to ECHAM-HAMMOZ aerosols (Figure S1). Aerosol pH distribution by AIM is used as input in the sensitivity simulation $\gamma$pH, while the reference simulation RefBase uses a uniform value for the reactive uptake coefficient $\gamma$ corresponding to a pH of around 2.5. The simulation $\gamma$pH was designed to to explore the impact of
5   such a dependence. Therefore, based on reaction probability values given in Eddingsaas et al. (2010) and Gaston et al. (2014) a simple function for $\gamma_{IEPOX}$ was formulated and implemented in ECHAM-HAMMOZ:

$$\gamma(\text{pH}) = \begin{cases} 10^{-2}, & \text{pH} < 2 \\ 0.1[\text{H}^+] + 10^{-4}, & \text{pH} \in [2,5] \\ 0, & \text{pH} > 5 \end{cases} \tag{2}$$

where $[\text{H}^+]$ is the concentration of protons $\text{H}^+$ in the aerosol given in $\text{mol}\,l^{-1}$. The reaction probability varies linearly between particles of pH values between 2 and 5. For acidic particles the upper limit of $10^{-2}$ is fixed. For particles which are
10   not acidic enough (pH greater 5) no reaction is assumed. The pH distribution (Figure S1) was then used as model input values. The pH value of the surface aerosols was applied to each model layer, but largest effect can be observed where acidic aerosol and LIEPOX are present.

    Figure 6 shows the resulting global surface distribution of $\gamma$pH run for northern hemispheric summer compared to RefBase. Enhancement of reactive uptake in $\gamma$pH over land is clearly visible, especially over Southeastern US maximum values are more
15   than doubled. Further, more areas with $3-4\,\mu g\,m^{-3}$ over Africa, the Middle East and Eurasia can be found, where RefBase has values lower than $1\,\mu g\,m^{-3}$. In contrast, suppression of LIEPOX reactive uptake is observable over the Amazon.

Total LIEPOX aerosol produced during this time period increased by 58 % in $\gamma$pH compared to RefBase. In RefBase an aerosol pH around 2.5 was assumed for all aerosols, also those which might be less acidic like sea salt aerosol. Nevertheless, compared to $\gamma$pH less LIEPOX-SOA was formed. In $\gamma$pH most areas are covered by less acidic aerosol but LIEPOX is produced or transported there, where acidic aerosol can be found, this leads to the observed increase in iSOA formation.

As an alternative explanation for the to pH value dependence, Xu et al. (2015) hypothesize that IEPOX uptake enhancement could be triggered by sulfate aerosol. Although sulfate aerosol is simulated no sensitivity study was performed here due to lack of process understanding and possible reactive uptake parametrisations.

### 3.2.3 Sensitivity to evaporation enthalpy

Tsigaridis and Kanakidou (2003) point out the sensitivity of SOA formation to the evaporation enthalpy $\Delta H_{vap}$. Nevertheless, due to the lack of knowledge of $\Delta H_{vap}$ for the different organic compounds, usually a fixed value or rather low value is used for all of them (Epstein et al., 2009). Depending on the study, different estimations for $\Delta H_{vap}$ were made, ranging between 30 and $156 \, \text{kJ} \, \text{mol}^{-1}$ (Athanasopoulou et al., 2012). Farina et al. (2010) also use the Clausius-Clapeyron equation to calculate saturation concentrations for a variety of organics using $30 \, \text{kJ} \, \text{mol}^{-1}$ as the $\Delta H_{vap}$. To explore the impact of this assumption and the impact of a lower evaporation enthalpy, the sensitivity run $\Delta$H30 was designed to use $\Delta H_{vap} = 30 \, \text{kJ} \, \text{mol}^{-1}$ but keeping the same reference saturation vapor pressure (see Table 1).

As an example Figure 7 shows the curves given by equation (1) using $\Delta H_{vap}$ of the reference run and the sensitivity run. Equation (1) changes its curve form drastically when lowering $\Delta H_{vap}$ from values around $150 \, \text{kJ} \, \text{mol}^{-1}$ to $30 \, \text{kJ} \, \text{mol}^{-1}$. For temperatures lower than the reference value of 298.15 K the saturation vapor pressure of $\Delta$H30 $p^*_{\Delta \text{H30}}$ is higher compared to the reference $p^*$, but for temperatures higher 298.15 K the opposite is the case (see Figure 7).

As a result, the impact of variable $\Delta H_{vap}$ on iSOA formation varies with temperature, therefore, also with region and height. The sensitivity simulation $\Delta$H30 ran for June, July and August 2012 with changed Clausius-Clapeyron equation curves according to Figure 7. Even during this northern hemispheric summer, on a global perspective the atmosphere is on average cooler than 298.15 K, especially at higher altitudes. Therefore, global total iSOA production in $\Delta$H30 for the considered time period is just 0.6 TgC lower compared to RefBase. This is a reduction of 4 % of the total amount produced in RefBase in June, July and August 2012. For surface temperatures higher than 298.15 K $p^*_{\Delta \text{H30}}$ is orders of magnitude lower than the reference $p^*$, but gas phase concentrations of iSOA precursors are high enough, that no significant impact on iSOA concentrations is seen. In agreement, surface concentration fields do not change much and are therefore not shown.

The assumption made by Farina et al. (2010) connected with the estimation of $p^*_0$ of iSOA precursors in this study therefore does not lead to significant changes in model results. Lowest sensitivity to $\Delta H_{vap}$ can be found in the LVOC, LISOPOOHOOH. In ECHAM-HAMMOZ sensitivity to $\Delta H_{vap}$ increases with volatility of the compounds, therefore $\Delta H_{vap}$ should be crucial for additional consideration of SVOC and IVOC, which will be added to the model in a future study.

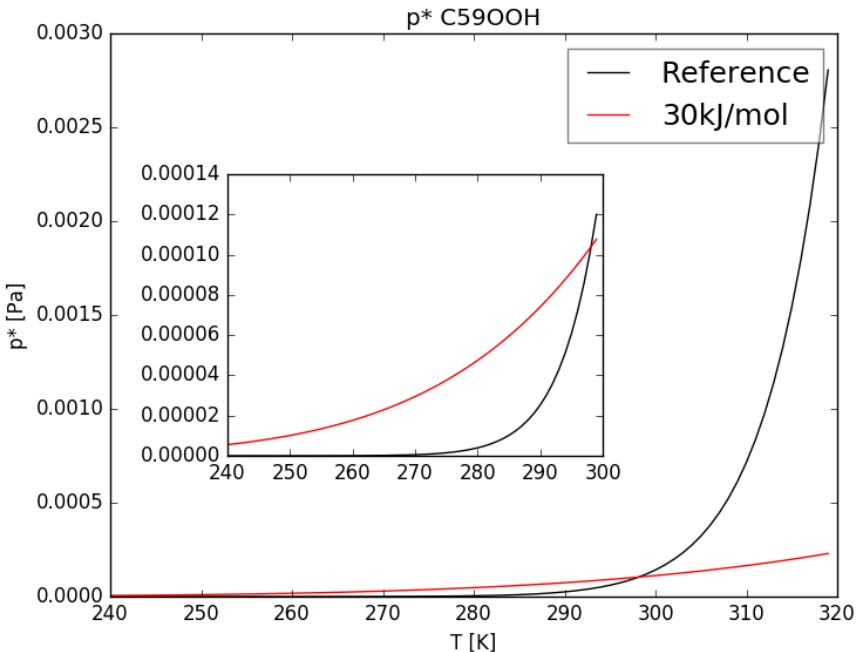

**Figure 7.** Curves given by Clausius Clapeyron equation 1 for C59OOH. The red curve is obtained by setting $\Delta H_{vap}$ = 30 kJ mol$^{-1}$, the black one describes the parameters used in the reference run (see Table 1).

### 3.2.4 Uncertainty estimation saturation vapor pressure

As described in Section 2.1.1 the group contribution method by Nannoolal et al. (2008) in combination with the boiling point method by Nannoolal et al. (2004) were used to obtain the saturation vapor pressure of originated isoprene products as a function of temperature. Group contribution methods estimate the contribution of functional groups on saturation vapor pressure. The Nannoolal et al. (2008) group contribution method is based on 68835 data points of 1663 components and just needs two inputs, the molecular structure and the normal boiling point. Nannoolal et al. (2008) report a good performance against measurements. Nevertheless, when its performance is compared to compounds outside the training set, results become worse (Barley and McFiggans, 2010; OMeara et al., 2014). Barley and McFiggans (2010) underline that databases are typically biased towards mono-functional groups and therefore, group contribution methods trained with these data perform well for volatile fluids, but not for low volatility compounds. OMeara et al. (2014) arrive at similar conclusions, they tested seven saturation vapor pressure estimation methods and found that even if Nannoolal et al. (2008) method results in the lowest mean bias error, the method shows poor accuracy for compounds with low volatility. This tendency also holds true for the other tested methods showing an increasing error with increasing number of hydrogen bonds. This systematic error results in a SOA formation overestimation.

**Table 5.** Comparison logarithmic saturation concentrations $\log_{10} C_0^*$ at 300 K for the iSOA precursors calculated via the group contribution method used in RefBase (Nannoolal et al., 2008), used in EVA (Compernolle et al., 2011) and a simple group contribution method formulated by Donahue et al. (2011). In brackets the $\log_{10} C_0^*$ for the isomers are shown.

|  | Nannoolal et al. (2008) | Compernolle et al. (2011) | Donahue et al. (2011) |
|---|---|---|---|
| LNISOOH | 1.2 (1.4) | 2.2 (2.8) | 1.3 |
| LISOPOOHOOH | -1.6 (-1.9) | -0.2 (-0.9) | -0.7 |
| LC578OOH | 1.1 (1.1) | 1.7 (1.7) | 1. |
| C59OOH | 0.8 | 1.4 | 1. |

Since often, the underlying databases used for group contribution methods, also for Nannoolal et al. (2008), are biased, not including complex polyhydroperoxides (Kurten et al., 2016), a sensitivity run with the group contribution method EVAPO-RATION was performed (EVA). EVAPORATION was designed to include hydroperoxides' and peracids' molecular structures and does not need a boiling point as an input (Compernolle et al., 2011). Especially for the highly oxidized compound LISOPOOHOOH this could reduce the model error.

Moreover, McFiggans et al. (2010) analyzed the dependence of SOA formation of the saturation vapor pressure of each compounds and state that SOA mass is highly sensitive to this parameter. Up to 30% overestimation can result from ignoring non-ideality of the organic mixture.

These studies already identified and emphasized several causes and consequences of the various group contribution methods. Thus, $\log_{10} C_0^*$ values by Nannoolal et al. (2008) method are compared to values obtained by the EVAPORATION method and a simple group method based on oxygen, carbon and nitrate atoms in the molecule described in Donahue et al. (2011) (Table 5).

As can be seen from Table 5, the $\log_{10} C_0^*$ values do not differ much between the simple group contribution method of Donahue et al. (2011) and the one by Nannoolal et al. (2008), except for the lowest volatility compound LISOPOOHOOH. For LISOPOOHOOH, Nannoolal et al. (2008) predict a much lower volatility than Donahue et al. (2011). This difference agrees with the findings of the studies described above and indicates that LISOPOOHOOH-iSOA formation might be too high in ECHAM-HAMMOZ. In contrast, the $\log_{10} C_0^*$ from the Nannoolal et al. (2008) method strongly differ from the $\log_{10} C_0^*$ calculated by the EVAPORATION method (Compernolle et al., 2011). All compounds are estimated to be more volatile when EVAPORATION is used. This changes the classification, according to the definitions by Donahue et al. (2012), of LISOPOOHOOH, which was referred as LVOC and would now be an SVOC ($\log_{10} C_0^* > -0.5$), and LNISOOH, which was called an SVOC, but now would rather be an IVOC (second isomer, $\log_{10} C_0^* > 2.5$).

Also given are the $\log_{10} C_0^*$ values for the different isomers. The Nannoolal et al. (2008) method and EVAPORATION method agree well in volatility of the isomers. Both calculate that the second LNISOOH isomer is more volatile, the second LISOPOOHOOH isomer is less volatile and that there is no difference between the LC578OOH isomers. Due to computational resource limits, no further sensitivity runs using, the isomers instead were done. Nevertheless, from the $\log_{10} C_0^*$ values and the values in Table 1 it is clear, that for LC578OOH there is no difference caused by isomeric structures in volatility, for LNISOOH the other isomer is even slightly more volatile and for LISOPOOHOOH the opposite holds true, its second isomer

**Table 6.** Percentage changes in total iSOA formation in 2012 for the sensitivity run EVA using EVAPORATION instead of Nannoolal et al. (2008) to estimate the saturation vapor pressure and the evaporation enthalpy of the isoprene derived SOA precursors.

| | Change in EVA compared to RefBase | |
| | % | TgC |
| --- | --- | --- |
| LIEPOX | -5.2 | -1.1 |
| IGLYOXAL | -5.2 | -0.2 |
| LISOPOOHOOH | -16.7 | -4.6 |
| C59OOH | -36.3 | -0.1 |
| LC578OOH | -50.0 | -0.3 |
| LNISOOH | -90.0 | -0.1 |
| Total iSOA | -12.8 | -7.3 |
| Total iSOA burden | -16.7 | -0.1 |

is slightly less volatile. Since LNISOOH is only formed in very low concentrations these deviations might not be visible in iSOA formation.

The large differences between Nannoolal et al. (2008) method and EVAPORATION method motivated the sensitivity run EVA, which was performed for one year to evaluate the iSOA budget. Higher volatility of iSOA precursors in EVA lead to less surface area available for reactive uptake of LIEPOX and IGLYOXAL. The changes in the production rates and iSOA burden can be found in Table 6 (Stadtler, 2018). Total iSOA production is reduced by 12.8 %, the iSOA burden by 16.7 %. This reduction is mainly explained by reduction in LISOPOOHOOH-SOA and LIEPOX-SOA formation, although LC578OOH-SOA and LNISOOH-SOA production is reduced by 50 % and 90 %. It should be noted that in EVA, most iSOA is produced by LISOPOOHOOH, as well, a total of 23.2 TgC, followed by 20.4 TgC by LIEPOX in 2012. Therefore, the iSOA composition is still dominated by these two compounds. Surface iSOA concentrations change marginally, the same patterns are visible as in RefBase and are therefore not shown. Thus, the main conclusions of this study, using the reference run RefBase, do not change although iSOA production is reduced when using the EVAPORATION method (Stadtler, 2018).

### 3.2.5 Uncertainty in LISOPOOHOOH aerosol

Sections 3.1.1 and 3.1.2 showed the large impact of LISOPOOHOOH on iSOA in ECHAM-HAMMOZ, but as seen in Section 3.2.4, the uncertainty in LISOPOOHOOH vapor pressure is high. The large contribution to iSOA remains, regardless of whether the Nannoolal et al. (2008) or EVAPORATION (Compernolle et al., 2011) method is used for saturation vapor pressure estimation. While it is difficult to conceive additional sources of LISOPOOHOOH, there are two pathways through which the impact of LISOPOOHOOH on particle phase concentrations could be lower than estimated here: (1) reduction of chemical LISOPOOHOOH production in the gas phase, (2) introduction of additional SOA sinks, e.g. in-particle decay and particle photolysis.

As pointed out in Section 2.1.1, the direct LISOPOOHOOH precursor, LISOPOOHO2, 1,5H-shift reaction is missing in the JAM3 chemical mechanism. To test the impact of this reaction, which leads to different iSOA precursors than LISOPOOHOOH, a simplified pseudo-reaction was added to the mechanism in the sensitivity experiments HshiftLIE and HshiftC5, respectively (Section 2.2). Both include the isomerization reaction of LISOPOOHO2, but yield different iSOA precursors. In HshiftLIE LISOPOOHO2 1,5H-shift produced LIEPOX, while in HshiftLC5 LC578OOH is produced instead. The real impact of the 1,5H-shift might lie between these runs, but they agree well with respect to the impact on total iSOA. Both runs diagnose a reduction in chemical LISOPOOHOOH production by 95 % and a reduction in LISOPOOHOOH-SOA formation by 92 %. Nevertheless, either LIEPOX-SOA or LC578OOH-SOA formation increases due to increased gas phase production of these compounds in each run respectively. IEPOX-SOA in HshiftLIE increases by 31 %, while there is 13 times more LC578OOH-SOA in HshiftLC5. Therefore, the iSOA burden is reduced in HshiftLIE by 37 % and in HshiftLC5 by 28 %. Global distributions for the simulated time period can be found in the Supplementary S2, Figures S2 and S3.

Concerning the potentially importaqnt additional iSOA sink processes, we tested particle decay with a simple parametrisation using a half-lifetime of 4 h following D'Ambro et al. (2017a) (SIMULATIONNAME). Hodzic et al. (2015) claim that SOA-photolysis might be a as high as $J_{SOA} = 0.04 \% J_{NO_2}$ and would thus constitute an efficient sink, which could lead to substantial reduction of SOA mass in the atmosphere. Malecha and Nizkorodov (2016) criticize this photolysis rate being too high. Following the simple scaling approach of Hodzic et al. (2015), we implemented iSOA photolysis with a rate of $J_{iSOA} = 0.004 \% J_{NO_2}$ and tested the impact of this reaction in simulation SIMULATIONNAME (Stadtler, 2018). Including both processes leads to a reduction of the iSOA burden by 50 %, whereby LISOPOOHOOH is reduced most effectively.

We conclude that there are indeed processes which have the potential to substantially reduce the contribution of LISOPOOHOOH to SOA formation and thus lower the iSOA burden compared to our base simulation by up to 50 %. However, quantification of the rates of these processes remains highly uncertain. Therefore they were not included in the base version of our chemical mechanism.

### 3.3 Comparison with observations

In order to evaluate how much of total organic aerosol (OA), including primary and secondary organic aerosol, are related to iSOA, iSOA concentrations and O:C ratios from ECHAM-HAMMOZ are compared to Atmospheric Mass Spectrometry (AMS) measurements from different field campaigns given in Table 7. Measurements were selected from the AMS global database (Zhang et al., last accessed on 22.09.2017) according to the availability of elemental ratios. All campaigns took place either in Europe or North America and include six different countries. In Helsinki, Finland winter and spring measurements are available.

Figure 8 shows the quartiles of the time series of the concentrations in the different locations, from left to right first the four European data sets and then the American ones. The European data sets display a variety of local OA sources. For Helsinki, Carbone et al. (2014) report a variety of local sources for OA including biomass burning, traffic, coffee roaster and also SOA from long range transport. In Mace Head two different OA types are measured depending on the advection of either marine air or continental air (Dall'Osto et al., 2009). Saarikoski et al. (2012) identified in Po Valley a complex mixture of OA was

**Table 7.** Overview ambient measurement locations, time periods and references. For Helsinki there are two time series, one during winter (W) and the second during spring (S).

| Location | Observation time period | Reference |
|---|---|---|
| Helsinki, Finland (60.2° N, 24.95° E) | 08.01. - 14.03.2009 (W) | Carbone et al. (2014) |
| | 09.04. - 08.05.2009 (S) | Timonen et al. (2010) |
| Mace Head, Ireland (53.33° N, 9.99° W) | 25.02. - 26.03.2009 | Dall'Osto et al. (2009) |
| Po Valley, Italy (44.65° N, 11.62° E) | 31.03. - 20.04.2008 | Saarikoski et al. (2012) |
| Houston, USA (29.8° N, 95.4° W) | 15.08. - 15.09.2000 | Zhang et al. (2007) |
| Mexico City, Mexico (19.48° N, 99.15° W) | 10.03. - 30.03.2006 | Aiken et al. (2009, 2010) |
| Manaus, Brazil (2.58° S, 60.2° W) | 06.02. - 13.03.2008 | Chen et al. (2009); Pöschl et al. (2010); Martin et al. (2010) |

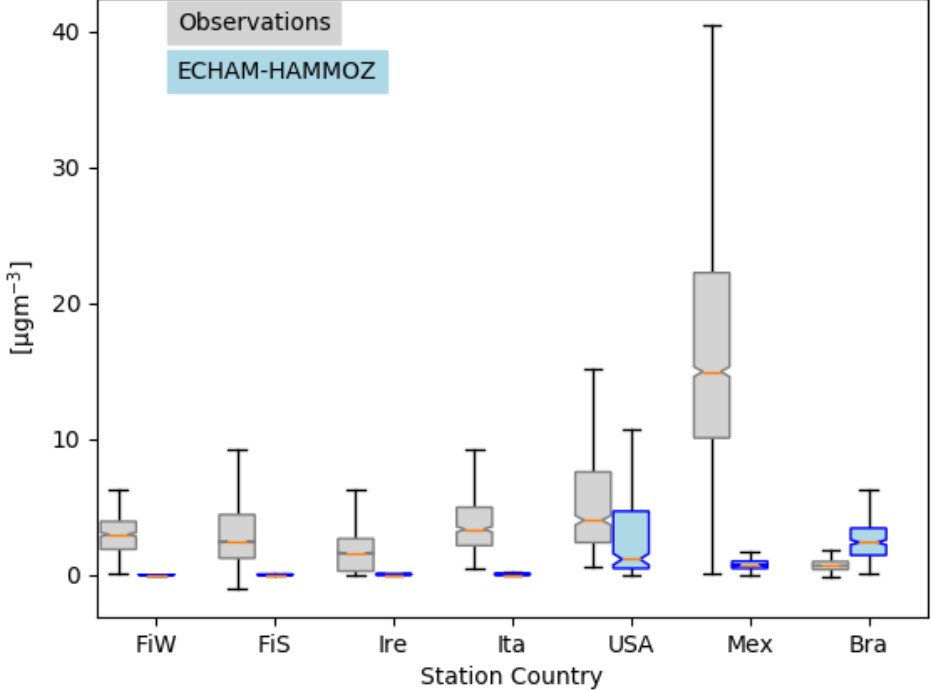

**Figure 8.** Box plots showing the variability of concentrations measured and corresponding instantaneous values from ECHAM-HAMMOZ. Countries of measurement campaigns are given. First, European countries then American ones. The shortcuts refer to: FiW=Helsinki, Finland (Winter), FiS=Helsinki, Finland (Spring), Ire=Mace Head, Ireland, Ita=Po Valley, Italy, USA=Houston Texas, USA, Mex=Mexico City, Mexico, Bra=Manaus, Brazil. The model time resolution is three hours, whereas all values given from the observations are included meaning that they have a higher time resolution.

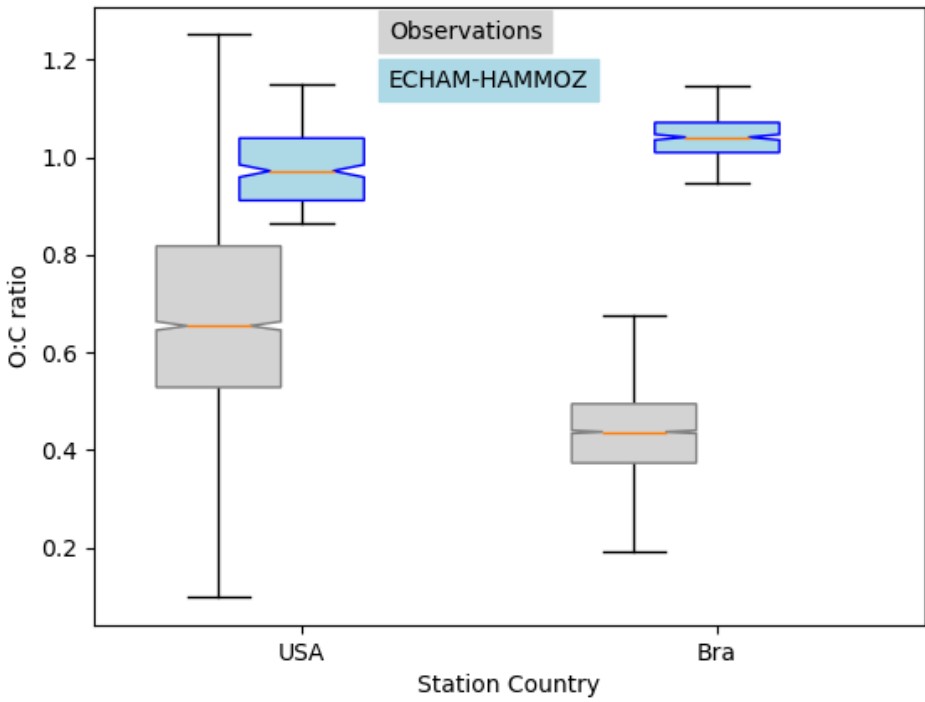

**Figure 9.** Similar to Figure 8 showing corresponding O:C ratios of the subset Houston Texas, USA and Manaus, Brazil. The O:C ratios shwon here are corrected by the factor 1.27 according to Canagaratna et al. (2015)

with local and regional sources, mainly from anthropogenic origin. For Finland, Ireland and Italy, ECHAM-HAMMOZ reveals a minor contribution of iSOA to OA this can be explained by the measurement time periods in winter or early spring where vegetation in Europe does not emit large isoprene amounts (Steinbrecher et al., 2009).

Looking at the concentrations measured in Houston Texas, USA it can be seen that a great part of the variability is cap-
5 tured by iSOA, which is explained by high isoprene emissions found in Southeastern US. ECHAM-HAMMOZ median and percentiles are still lower than the observations since the observation includes total OA. The organic aerosol in Mexico City was measured at an urban super-site and covers such a big range of concentrations, which are dominated by anthropogenic emissions including biomass burning, nitrogen containing OA and primary hydrocarbon like OA associated with traffic (Aiken et al., 2009). According to the concentrations simulated by ECHAM-HAMMOZ, just a minor part of these can be explained by
10 iSOA. Manaus, Brazil is located in the Amazon Basin and classified as pristine environment close to pre-industrial conditions (Pöschl et al., 2010; Martin et al., 2010). Therefore, the particles are nearly purely biogenic and Martin et al. (2010) report an upper limit of 5% primary organic aerosol. These conditions are ideal to compare them to ECHAM-HAMMOZ just including iSOA, because isoprene emissions are high in the Amazon Basin and should dominate the OA there. As can be seen in Figure 8,

HAMMOZ simulates overall higher iSOA concentrations that OA concentrations measured. Moreover, higher peak values are simulated and the median is higher than the upper 1.5 inter-quartile range whisker of observed concentrations.

For Houston Texas and Manaus, the comparison of the concentrations simulated by ECHAM-HAMMOZ and the concentrations measured by the AMS shows that ECHAM-HAMMOZ relates a great part of the observed OA to iSOA. Since iSOA seems to play a role in these regions, O:C ratios are compared as well. Due to restricted iSOA formation in ECHAM-HAMMOZ only from six iSOA precursors which are highly oxidized molecules with molecular O:C ratios between 0.6 and 1.4, the modeled O:C ratio just covers small variability and is around 1 in both regions, see Figure 9.

The comparison of concentration spectrum in Houston Texas showed a great part to be attributed to iSOA, this modeled subset covers upper values of the O:C ratio between 0.8 and 1.1, which still lie within the 75[th] percentile and the upper 1.5×IQR whisker of the measured data. This is related to the fact of missing SVOC and IVOC usually having lower O:C ratios and the contribution of POA to OA, which is not included in this comparison, because no assumptions of POA O:C ratios are made.

In contrast, the OA measured in Manaus located at the Amazon Basin, which consists of 95 % SOA does not show as high O:C ratios as iSOA modeled by ECHAM-HAMMOZ. The median of observed aerosol lies at 0.4, instead of 1. Certainly part of it is explained by missing SVOC and IVOC in ECHAM-HAMMOZ, but might also be related to SOA from other organic molecules than isoprene.

For Manaus an overestimation of iSOA concentrations by the model might be related to mistakes in emissions and in the chemical mechanism, missing sink processes and uncertainties in $p^*$. In term of O:C ratio of modeled iSOA between 0.6 and 1.4, the simulated values are covered by the ambient values in Houston Texas, but not in Manaus. This points to SVOC, IVOC and SOA from other sources than isoprene.

To summarize, isoprene emissions are not dominating OA in Europe, therefore the model shows iSOA having a small contribution to concentrations there. In contrast, American OA is more impacted by iSOA, especially in USA and Brazil.

## 4 Discussion

The comparison of RefBase to the AeroCom models, ECHAM-HAM and AMS measurements in the isoprene dominated area Manaus in the Amazon basin revealed that semi-explicit treatment of atmospheric chemistry, at least for isoprene, leads to much larger SOA production rates and eliminates low biases found in most other global model studies. In fact, especially over Brazil, SOA now has a tendency to be overestimated. Extrapolating iSOA production rate to the production rate of SOA in ECHAM-HAMMOZ including SVOC, IVOC not only from isoprene, but also from terpenes and aromatics, we expect to find a portion of SOA which cannot be reduced by only including the missing aerosol sinks. Various reasons for this part of the overestimation of iSOA in ECHAM-HAMMOZ could be identified, analyzing the results of the simulations presented in this study and are discussed in the following.

First, overestimation of iSOA may be related to errors in the group contribution method used to estimate the saturation vapor pressure and evaporation enthalpy of iSOA precursors. As discussed in OMeara et al. (2014) and Barley and McFiggans (2010)

the Nannoolal et al. (2008) method is problematic in the low volatility range, giving too low saturation pressures, which leads to an overestimation in SOA formation. Comparing the logarithm of the saturation concentration at a reference temperature ($\log_{10}C_0^*$) obtained by Nannoolal et al. (2008) to the one using the EVAPORATION (Compernolle et al., 2011) method and to the simple method of Donahue et al. (2011) reveals greatest differences in the lowest volatile compound LISOPOOHOOH.

To assess the impact of a reduced $\log_{10}C_0^*$, a sensitivity simulation using the EVAPORATION method was performed. This simulation showed a reduction in the global, annual, average iSOA burden of 16.7 %, which is mainly due to reductions in LISOPOOHOOH-SOA. Nevertheless, the main conclusions obtained from the reference run are not changed by this sensitivity simulation.

    A second aspect leading to a potential overestimation of iSOA is the semi-explicit chemistry itself. Different chemical

pathways lead to formation of isoprene SVOCs and LVOCs, some requiring NO and $NO_3$ for the initial steps followed by OH, $HO_2$ or $RO_2$. Formation of iSOA precursors via the $NO_x$ depended pathways hardly happens, as can be seen in the chemical budget terms. LNISOOH and LC578OOH are formed in very low concentrations and C59OOH might just result from the $HO_2$-dominated pathway. From the chemical branching it can be clearly seen that on average in JAM3 the OH initiated pathway is preferred, even in regions were NO mixing ratios are higher than 200 pptv (not shown). 90% of iSOA consist

of products from this pathway, mostly IEPOX and LISOPOOHOOH. Highly acidic aerosol is expected in regions where sulfate pollution is high and these regions usually coincide with high $NO_x$, which should suppress LIEPOX formation. For LIEPOX this might lead to a large overestimation when acidic enhancement is considered. The JAM3 chemical mechanism simulates LIEPOX and LISOPOOHOOH suppression when NO mixing ratios surpass 700 pptv. This feature in the chemistry can only be seen in 3 hourly values of the single grid boxes and vanishes when mean values are evaluated. Once NO mixing

ratios are lower, LIEPOX and LISOPOOHOOH are produced again, leading on average to the impression of missing $NO_x$ suppression (Stadtler, 2018). Further, LISPOOHOOH production might be overestimated due to a missing intramolecular 1,5H-shift of LISOPOOHO2 in JAM3, which would lead to products with a saturation vapor pressure which is around 2 orders of magnitude higher than the one of LISPOOHOOH (see Section 2.1.1, D'Ambro et al. (2017b)). To check the impact on LISOPOOHOOH and the derived iSOA, two tests including the 1,5H-shift of LISOPOOHO2 were evaluated. Gas phase

production of LISOPOOHOOH is reduced by over 90 %, but since the products of 1,5H-shift still form iSOA, iSOA burden is only reduced by around 30 %.

    Third, the main iSOA formation pathways follow from OH initiated reactions, which is the main oxidation pathway for isoprene. The ECHAM-HAMMOZ evaluation of Schultz et al. (2017) shows that the tropical region in our model is too wet. This leads to a higher production of OH radicals so that the model atmosphere is more oxidative than the real atmosphere.

Tropical regions are those, where most of isoprene is emitted. Thus, gas phase precursor formation might be overestimated already, which translates into an iSOA overestimation.

    Fourth, Kroll et al. (2006) reported rapid chemical loss of SOA via photolysis could be a possibility to further transform iSOA either to higher oxidized molecules in the particle phase such as oligomers, or to fragment iSOA compounds leading to VOC and iSOA reduction. Hodzic et al. (2015) explored the global impact of SOA photolysis and report about a $40-60\%$

mass reduction after 10 days. SOA photolysis is closely related to wet-phase, in-particle chemistry, which is not included in the

ECHAM-HAMMOZ chemical mechanism. Thus, simple paramterisations for SOA photolysis and in-particle decay were tested with ECHAM-HAMMOZ, efficiently reducing the iSOA burden by around 50 % due to in-particle decay of LISOPOOHOOH and by around 10 % due to SOA photolysis.

Finally, model limitations in aerosol and cloud processing did not allow to implement in-cloud iSOA formation. This is not only a potential additional source, but also an additional sink. ECHAM-HAMMOZ just includes wet scavenging based on solubility following Henry's law, but according to Cole-Filipiak et al. (2010) the IEPOX hydrolysis reactions at low pH values have lifetimes comparable to wet deposition. Heterogeneous uptake of IEPOX in cloud droplets and rain would lead to a decrease in gas phase concentrations while not resulting in iSOA, because it is lost immediately due to precipitation. This would lower iSOA from LIEPOX, which now has a substantial contribution to total iSOA.

# 5    Conclusions

For the first time, the semi-explicit chemical treatment of isoprene oxidation in the chemical mechanism of a global chemistry climate model was connected to explicit partitioning of individual low volatility species according to their chemical structures. The chemistry model MOZ includes a total of 779 reactions, where 147 reactions describe the isoprene oxidation. Isoprene oxidation in MOZ leads to iSOA precursors which are explicitly partitioned and followed in specific aerosol bins by HAM-SALSA. The partitioning is based on the saturation vapor pressure derived from the molecular structure of each single compound. Furthermore, also reactive uptake of isoprene derived glyoxal and IEPOX was considered.

These two iSOA formation pathways lead to a global, annual, average isoprene SOA yield of 15 % relative to the primary oxidation of isoprene by OH, $NO_3$, and ozone in 2012. It was identified that in ECHAM-HAMMOZ most iSOA is produced via the OH oxidation initiated pathway which leads to production of IEPOX and ISOP(OOH)2, a compound recently detected in experimental studies. Together modeled IEPOX and ISOP(OOH)2 yield a fraction of 90 % of total iSOA mass. In total 56.7 TgC iSOA are produced. IEPOX forms 21.0 TgC and ISOP(OOH)2 27.9 TgC aerosol. 54.7 TgC iSOA are lost due to wet deposition, which is the main sink for iSOA in ECHAM-HAMMOZ. For 2012 an average iSOA burden of 0.6 TgC is calculated. These values were compared to SOA budgets in AeroCom models. ECHAM-HAMMOZ simulates a higher production rate than all models used in this AeroCom study.

Moreover, this explicit model system enables process understanding and discussion. While exploring the influence of aerosol pH on IEPOX reactive uptake, enhancement of iSOA formation was found especially over Southeastern US while suppression could be observed over the Amazon basin.

Evaporation enthalpy used in previous model studies was compared to the explicitly derived evaporation enthalpy used in ECHAM-HAMMOZ. A huge difference could be found in Clausius-Clapeyron curves, which does not translate to a big impact on iSOA formation due to the fact that only sufficiently low volatile precursor gases were used.

Changing the volatility of the partitioning precursor gases has a larger impact on iSOA formation and the SOA burden. The group contribution methods by Nannoolal et al. (2008) and Compernolle et al. (2011) were used and annual iSOA formation compared. EVAPORATION estimates higher saturation vapor pressures for all partitioning precursors, leading to a 16.7 %

reduction in global, annual, average iSOA burden. Nevertheless, the iSOA composition does not change: LISOPOOHOOH and LIEPOX still are the dominating compounds.

LISOPOOHOOH-SOA formation can only be reduced by either including new iSOA sinks or the isomerization reaction of the direct LISOPOOHOOH precursor. This 1,5H-shift reaction still leads to iSOA precursors, thus LISOPOOHOOH-SOA is reduced largely by 90 %, but total iSOA burden only by 30 %, because the new iSOA precursor still contributes to iSOA.

Comparison of ECHAM-HAMMOZ iSOA concentrations to AMS measurements showed that ECHAM-HAMMOZ does not underestimate iSOA formation. On the contrary, in isoprene dominated regions like Brazil an iSOA overestimation could be observed which gives the possibility to explore novel SOA sinks, like in-particle decomposition or photolysis. Not lowering the production rate, but including additional sinks is the strategy proposed by Hodzic et al. (2016). They conclude that because several SOA sinks are currently excluded from global models, these models are expected to overestimate SOA concentrations. Instead of the expected overestimation, an underestimation is found in the majority of global models. With our explicit model system connecting the aerosol bin scheme SALSA with the chemistry model MOZ in the frame of the global model ECHAM-HAMMOZ sufficient SOA is produced to be able to explore new sink processes. The simple implementation and quick test of in-particle decay and SOA photolysis reduced the global iSOA burden about 50 %.

*Code availability.* The ECHAM6.3-HAM2.3MOZ1.0 model code can be found at https://redmine.hammoz.ethz.ch/projects/hammoz/wiki/ Echam630-ham23-moz10. After registration the code is available for download via the Apache Subversion system (SVN). All changes made according to SALSA and MOZ coupling can also be found there in the Finnish Meteorological Institute (FMI) branch. Please do not hesitate to ask the authors for support getting the code.

*Competing interests.* No competing interests are present.

*Author contributions.* SS and TK developed the coupling of HAM-SALSA and MOZ. For this TK adjusted the SALSA code to be compatible with MOZ, accordingly SS worked on the MOZ code to introduce tracer characteristics (saturation vapor pressure, reactive uptake etc.) needed by SALSA. The experiments were planned by SS, MS, DT, TK and HK. SS carried out all simulations, including the ECHAM-HAM simulation. SS prepared the manuscript with contributions from all co-authors.

*Acknowledgements.* ECHAM-HAMMOZ simulations were supported by the Forschungszentrum Jülich and performed at the Jülich Super-computing Centre (2016). The ECHAM-HAMMOZ model is developed by a consortium composed of ETH Zurich, Max Planck Institut für Meteorologie, Forschungszentrum Jülich, University of Oxford, the Finnish Meteorological Institute and the Leibniz Institute for Tropospheric Research, and managed by the Center for Climate Systems Modeling (C2SM) at ETH Zurich. Moreover, we acknowledge the Academy of Finland project no. 308292 and 307331, Nordforsk project no. 57001. We also want to thank Q. Zhang, C. Parworth, M. Lech-

ner, and J.L. Jimenez for facilitating the comparison with observations with their AMS global database. K. Tsigaridis is acknowledged for sending detailed results from the AeroCom model inter-comparison. Further, we acknowledge the authors involved in measuring organic aerosol in the different campaigns shown in this study.

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
