# Peer review of "Isoprene derived secondary organic aerosol in the global aerosol-chemistry-climate model ECHAM6.3.0-HAM2.3-MOZ1.0"

_Geoscientific Model Development, 2017_

## Short Comment (SC1) · 7 Nov 2017

Dear authors,

in my role as Executive editor of GMD, I would like to bring to your attention our Editorial version 1.1:

http://www.geosci-model-dev.net/8/3487/2015/gmd-8-3487-2015.html

This highlights some requirements of papers published in GMD, which is also available on the GMD website in the 'Manuscript Types' section:

http://www.geoscientific-model-development.net/submission/manuscript_types.html

In particular, please note that for your paper, the following requirements have not been

met in the Discussions paper:

- "The main paper must give the model name and version number (or other unique identifier) in the title."

Thus please add the model name (and/or its acronym) and the version number in the title of your article in your revised submission to GMD. E.g the title could be., "Isoprene derived secondary organic aerosol in the global aerosol chemistry climate model ECHAM(v6.3)-HAM(v2.3)MOZ(v1.0) "

Yours,

Astrid Kerkweg

---

## Referee Comment (RC1) · Anonymous Referee #1 · 27 Nov 2017

Review of the manuscript "Isoprene derived secondary organic aerosol in a global aerosol chemistry climate model"

In the manuscript the authors describes a semi-explicit isoprene oxidation scheme and the explicit treatment of SOA formation from 6 isoprene oxidation products in the global chemistry climate model ECHAM-HAMMOZ.

I find the method used to simulate isoprene SOA (iSOA) very novel and encouraging and I definitely think that this kind of explicit treatment of SOA formation from known oxidation products are necessary in order to improve the knowledge about SOA formation and anthropogenic impact (e.g. NOx) on SOA formation in the atmosphere.  The manuscript is generally well written and clearly structured and I recommend that the manuscript should be accepted for publication after a minor revision where the authors carefully addressed my comments.

**General comments:**

I miss references in the main text to the supplement and the reactions R1-R22. This is needed for clarity.

On Page 5, L9 and on several other places in the manuscript you refer to the isoprene oxidation products in Table 1 as low volatility compounds (LVOC) since their saturation vapor pressures ($p_0$*) at 298.15 K is below 0.01 Pa. I presume that you mean their pure-liquid saturation vapor pressures. Further, this is not the common $p_0$* limit for LVOC which e.g. according to Donaue et al., Faraday Discuss. 2013, 165, 91–104 is compounds with C* in the range of $10^{-0.5}$ to $10^{-3.5}$ µg m$^{-3}$ which approximately correspond to a $p_0$* range of 4x10$^{-6}$ to 4x10$^{-9}$ Pa. Instead 0.01 Pa is somewhere on the limit between SVOC and IVOC. Thus, only LISOPOOHOOH in Table 1 can be referred to as LVOC at 298.15 K.

On Page 6-7 and in Figure 1 you refer to the yields of the iSOA precursors as if they were fixed yields. E.g. On Page 6, L9-10 you write that:  "LISOPOOHOOH has the highest yield of the LVOC considered here, 9% of isoprene end up as LISOPOOHOOH".
First of all, what do you mean with this yield? Is it the molar or mass yield that is 9 %?
Secondly, how can these yields have a fixed value? If the gas-phase mechanism explicitly simulates the formation of the different iSOA precursors their yields should vary depending on the concentrations of NO, O$_3$, OH, NO$_3$ etc. Maybe it is the global average yields derived with the model? If this is the case please state that.

 The section 2.1.2 is somewhat confusing when you discuss the SALSA-VBS system that apparently is not used in the present study but then in Figure 5 you still present results where you use the VBS method.

On Page 10, L12-13 you write: "Loss processes for SOA in HAM-SALSA include sedimentation, deposition and wash out in the aerosol phase." But should it not

be "Loss processes for SOA in HAM-SALSA include, apart from evaporation, sedimentation, deposition and wash out in the aerosol phase."
I don't understand why/how the pH of the aerosol particles can differ by several order of magnitudes if the only inorganic species is ammonium bisulfate. I guess the pH is only calculated for the inorganic aqueous phase. Is the 4 order of magnitude difference in pH only related to the average RH at different locations? Please clarify this in the text on page 10 and also refer to the figure in the supplement.

I miss labeling (e.g. a, b, c ) of the panels in all figure and the figure text to figure 2 need to describe what the two panels actually display.

In the results section on Page 14 and maybe on some other places you use the terms "quite big" and "quite high". Try to avoid using these vague statements and instead write concrete numbers (e.g. contributes to 30 % of the SOA mass instead of contributes to quite big amounts of the SOA mass).

On Page 14, L10 you state that LSOPOOHOOH-SOA is extremely low volatile but it is not. At least not at 298.15 K (see the comment concerning LVOC above).

Concerning the uncertainty of the estimated saturation vapor pressures (Section 3.3). I think it is good that you discuss the uncertainties related to the functional group contribution methods used to estimate the pure liquid saturation vapor pressures.
In publication by Kurtén et al., J. Phys. Chem. A 2016, 120, 2569–2582 it is stated that: "Unfortunately, the basis data sets of existing group contribution based empirical parametrizations for determining saturation vapor pressures of organic molecules (e.g., SIMPOL, or the widely used and generally successful Nannoolal et al. approach) do not contain complex polyhydroperoxides (or multiply substituted peroxy acids), and the parametrizations may therefore not be reliable for HOMs." LSOPOOHOOH can be considered as a HOM with two hydroperoxide functional groups.
Thus, according to the results from Kurtén et al., 2016 it may have been more appropriate to use the EVAPORATION method which also is provided via UManSysProp (Topping et al., 2016). With EVAPORATION/Nannoolal I get a $p_0^*$ (298.15 K) of $10^{-5}$ Pa, e.g. 38 times higher than the value you used at 298.15 K. In fact according to EVAPORATION LSOPOOHOOH is not a LVOC but a SVOC ($C^* \approx 1$ µg m$^{-3}$) at 298.15 K.

**Technical corrections:**
Write out that ISOP(OOH)2 refers to dihydroxy dihydroperoxide in the abstract

Page 4, L16: Change from "… and its evaluation it is referred …" to e.g. "… and its evaluation the reader is referred …"

Page 4, L33: Change from "The semi-explicit isoprene oxidation with 147 reactions constitutes a major of these reactions in JAM3."
e.g. to:
"Thus, the 147 reactions in the semi-explicit isoprene oxidation scheme

constitute a substantial fraction of these reactions in JAM3."

Page 5, L20: Change from " For simplicity there are …" to " For simplicity they are …"

On Page 15, L24: I think it should be "lifetime" and not "life time". Try to reformulate the wording "huge wet deposition loss".

On Page 17, L3: Remove "is" after iSOA and remove one "motivated" on L5.

On Page 17, L14: What do you mean with VBS classes 0, 1 and 10? If it is C* in µg m$^{-3}$ non of these species are LVOC at 298.15 K according to e.g. Donahue et al., 2013.

---

## Referee Comment (RC2) · Anonymous Referee #2 · 20 Feb 2018

Review of "**Isoprene derived secondary organic aerosol in a global aerosol chemistry climate model**"

By Stadtler *et al*. for *Geosci. Model Dev. Discussion*

General Comments

In this manuscript, the authors have developed the global chemistry climate model ECHAM-HAMMOZ to include a more explicit coupling between the gas and particle phase models in order to describe the formation isoprene derived secondary organic aerosol (SOA).  With their model, they predict that most of the iSOA is produced by IEPOX and $Isop(OOH)_2$.  This ultimately leads to over predictions of iSOA in relatively pristine locations where models typically under predict SOA in general.  There seems to be a growing tendency in the literature for models to capture ever increasing complexity in the chemical mechanisms because they are more capable of describing the wide variance of atmospheric conditions. For this reason, I believe this manuscript has a lot of value to interested readers.  However, it is incumbent on the developers to describe in detail all the relevant additions to the mechanism and justify other aspects that were not considered.  I feel that certain aspects of the chemical mechanism were not adequately characterized in this manuscript.  For example, there was no discussion on $HO_x$ recycling in the mechanism (an important facet in low $NO_x$ regimes) or how specifically all the percentage yields in Figure 1 were obtained – both of which affect oxidation state, product yields and branching ratios – and therefore model results.  For this reason, I would reconsider this publication after addressing the major and minor revisions detailed below.

Specific Comments

1.  The chemical mechanism as shown in Figure 1 contains many percentage yields.  The authors described the reaction pathways and mentioned yields in the text on pages 6 and 7 although they either did not provide references or a brief discussion of how the yields were obtained.  It may be stated in another reference but the crucial reaction yields shown in the figure need to be justified.  For example, how was the 9% gas phase yield of LISOPOOHOOH obtained or how was the 1% gas phase yield of LC578OOH determined?  It is these numbers which will directly affect SOA yields and it is therefore crucial to understand their uncertainties based on how they were derived.  A discussion with pertinent references should be included on pages 6 and 7.

2.  $HO_x$ recycling remains an issue in atmospheric chemistry models because $HO_x$ levels are typically under predicted in areas of low $NO_x$.[Archibald et al., "Impacts of HOx regeneration and recycling in the oxidation of isoprene: Consequences for the composition of past, present and future atmospheres", *Geophys. Res. Letters*, 2011, L05804.]  Certain reactions will rapidly consume $HO_x$ such as the formation of LISOPOOHOOH (2 OH and 2 $HO_2$ radicals typically consumed) while other reactions will recycle $HO_x$ such as the ring closure reaction of the $IsopOOH-(OH)_2$ radical to form IEPOX or intramolecular hydrogen shift reactions.  The consumption of $HO_x$ species has been expressed in the R1-R22 reactions but there seems to be no mention of $HO_x$ species regeneration which affects the oxidative capacity of the atmosphere in regions of low $NO_x$.  For example, reaction R3 will release OH radicals when IEPOX is produced but this is not specified in the reaction.  A hydrogen shift reaction (not really discussed in any of the reactions as far as I can tell) may produce carbon centered radicals at hydroxyl sites that may react with $O_2$ to yield a carbonyl compound and $HO_2$.  These regenerated $HO_x$ species are

important and need to be accounted for and/or discussed in the paper in the section describing these reactions.

3. On page 6 of the manuscript (line 7) it states: "Not included is the H-shift of LISOPOOHO2 that yields much more volatile compounds than LISOPOOHOOH". I do not agree with the authors that the compounds produced would be 'much more volatile' and therefore are not relevant to particle phase partitioning. If a 1,5-H-shift occurs in LISOPOOHO2, it would lead to a compound similar to LC578OOH except it would be heavier by one oxygen atom (i.e. a hydroxy-dihydroperoxy carbonyl derivative instead of the LC578OOH diol). Because LC578OOH partitions to the particle phase, so too would this newly produced compound derived from an H-atom shift. This product is indeed less volatile than LISOPOOHOOH, but it would be expected to partition into a particle phase thereby decreasing the influence of LISOPOOHOOH in the mechanism.

4. The product branching ratios for the subsequent reactions of IsopO2 in Figure 1 seem fixed regardless of the environment. Is this the intended assumption? Because all the subsequent reactions of IsopO2 are bimolecular, the branching ratios (and therefore product yields) will depend on the relative concentrations of $RO_2$, $HO_2$ and NO radicals. The gas phase product yields will therefore not only be influenced by local isoprene concentrations but also on the relative concentrations of these radicals. A discussion of this effect should be included in the manuscript along with a justification as to why using these fixed values represents an average isoprene environment.

5. The acronyms used to describe the chemical mechanism are not very clear. For instance, I cannot figure out what IsopOH is. I presume that IsopOOH is a hydroxy-hydroperoxy isoprene species (of which there are 8 isomers) so does that mean IsopOH is the diol? Chemical structures for all species listed in Table 1 and Figure 1 would be extremely useful.

6. In the chemical mechanism, there is no mention of dinitrate formation which is likely to occur in high $NO_x$ environments.[see Piletic et al. "Barrierless Reactions with Loose Transition States Govern the Yields and Lifetimes of Organic Nitrates Derived from Isoprene", J. Phys. Chem. A, 2017, 8306 and Jenkin et al. "The MCM v3.3.1 degradation scheme for isoprene", Atmos. Chem. Phys., 2015, 11433.] These species are highly oxidized and relatively heavy and therefore may affect the SOA yield in high $NO_x$ regime.

Technical Comments

1. On page 3 line 21, remove "the" for "In the light of …"
2. On page 5 line 14, replace "…., it is referred to the …" with "…, the reader is referred to the …"
3. On page 5 line 17, replace "Also the O3 initiated reactions pathways are included in MOZ, but none of the products was low volatile enough." with "The O3 initiated reaction pathways are included in MOZ, but the products are too volatile to contribute to SOA."
4. On page 9 line 17, add a comma between the words "dependence" and "sensitivity".
5. On page 9 line 25, replace "…processes to only the aerosol sized that are relevant…" with "…processes to include only the aerosol sizes that are…"
6. On page 9 line 30, replace "Here this model…" with "Here, the model…"

7. On page 12 line 24, replace "… especially LISOPOOHOOH molar mass of 168.14 g/mol is very large." with "…especially due to LISOPOOHOOH which has a molar mass of 168.14 g/mol that is very large."
8. On page 14 line 2, replace "… 2-methyltetrols in the order of ng/m3 are measured in…" with "… 2-methyltetrols are present in ng/m3 concentrations in the…"
9. On page 14 lines 9 and 18, the sentences are poorly expressed and need to be rewritten (i.e. "On the LISOPOOHOOH-SOA plot…" and "Hodzic et al…"
10. On page 14 line 33, the sentence should read "24% of isoprene ends up as IEPOX, 9% as LISOPOOHOOH, …" where every 'in' is replaced with 'as'
11. On page 15 line 7, it should read "The majority of precursors are destroyed chemically …"
12. On page 15 line 19, replace "… AeroCom mean value, because iSOA…" with "… AeroCom mean value because the iSOA…"
13. On page 17 line 5, remove "motivated" (word duplicated)
14. On page 17 line 21, replace "in contrary as can be seen in Figure 5 iSOA…" with "On the contrary as can be seen in Figure 5, iSOA…"
15. On page 20 line 14, replace "…of the various different organic …" with "…for the different organic compounds…"
16. On page 20 line 17, replace "…using for all of them 30 kJ/mol." with "using 30 kJ/mol as the $\Delta H_{vap}$."
17. On page 21 line 11, replace the word 'at' with 'for' at the end of the line.
18. On page 21 line 14, switch the order of "holds also" to "also holds".
19. On page 22 line 20, remove the word 'atmospheric' (redundant).
20. On page 22 line 21, add the word 'the' before AMS at the end of the line.
21. On page 25 line 3, replace the word 'pure' with 'purely'.
22. On page 25 line 8, the sentence either needs to be split up or more clearly stated.
23. On page 26 line 12, add the word 'the' between 'has most'
24. On page 26 line 16, add the word 'by' between 'followed OH'
25. On page 26 line 17, the sentence should read "… NOx dependent pathways…"

---

## Author Comment (AC1) · 23 May 2018

**Reply to executive editor comment on "Isoprene derived secondary organic aerosol in a global aerosol chemistry climate model"**

"The main paper must give the model name and version number (or other unique identifier) in the title."

Reply: Dear Astrid Kerkweg, thank you for the comment, we added the model name and version numbers to the title.

---

## Author Comment (AC2) · 23 May 2018

**Replies to Interactive comments on "Isoprene derived secondary organic aerosol in a global aerosol chemistry climate model" by Scarlet Stadtler et al.**

**1 Anonymous Referee #1**

In the manuscript the authors describes a semi-explicit isoprene oxidation scheme and the explicit treatment of SOA formation from 6 isoprene oxidation products in the global chemistry climate model ECHAM-HAMMOZ. I find the method used to simulate isoprene SOA (iSOA) very novel and encouraging and I definitely think that this kind of explicit treatment of SOA formation from known oxidation products are necessary in order to improve the knowledge about SOA formation and anthropogenic impact (e.g. NOx) on SOA formation in the atmosphere. The manuscript is generally well written and clearly structured and I recommend that the manuscript should be accepted for publication after a minor revision where the authors carefully addressed my comments.

Reply: We thank the referee for the encouraging, introducing words and for the following, very useful comments and corrections. The main replies are listed below according to the comments.

**1.1 General comments**

I miss references in the main text to the supplement and the reactions R1-R22. This is needed for clarity.

Reply: The references to the chemical reactions in the main text were added. Moreover, according to the comments of Reviewer #2 the reaction equations were adjusted to show also radical turnover.

On Page 5, L9 and on several other places in the manuscript you refer to the isoprene oxidation products in Table 1 as low volatility compounds (LVOC) since their saturation vapor pressures ($p_0$*) at 298.15 K is below 0.01 Pa. I presume that you mean their pure-liquid saturation vapor pressures. Further, this is not the common $p_0$* limit for LVOC which e.g. according to Donaue et al., Faraday Discuss. 2013, 165, 91–104 is compounds with C* in the range of 10-0.5 to 10-3.5 $\mu g\,m^{-3}$ which approximately correspond to a $p_0$* range of $4\times10^{-6}$ to $4\times10^{-9}$ Pa. Instead 0.01 Pa is somewhere on the limit between SVOC and IVOC. Thus, only LISOPOOHOOH in Table 1 can be referred to as LVOC at 298.15 K.

Reply: Indeed, the classification of the compounds according to Donahue et al. 2013 fits better to current literature and was adjusted thought the text. Especially, after running the sensitivity simulation using EVAPORATION instead of Nannoolal et al. 2008, the classification of

the four compounds as SVOC or LVOC was further complicated. Table 5 was extended and with the EVAPORATION values for $\log 10(C_0^*)$, for example, pushing LNISOOH ($\log 10(C_0^*) = 2.2$) to the group of IVOC and even LISOPOOHOOH gets close to be classified as an SVOC. This is further discussed in Section 3.3.

On Page 6-7 and in Figure 1 you refer to the yields of the iSOA precursors as if they were fixed yields. E.g. On Page 6, L9-10 you write that: "LISOPOOHOOH has the highest yield of the LVOC considered here, 9 % of isoprene end up as LISOPOOHOOH". First of all, what do you mean with this yield? Is it the molar or mass yield that is 9 %?

Secondly, how can these yields have a fixed value? If the gas-phase mechanism explicitly simulates the formation of the different iSOA precursors their yields should vary depending on the concentrations of NO, $O_3$, OH, NO3 etc. Maybe it is the global average yields derived with the model? If this is the case please state that.

Reply: The text was misleading, indeed there are no fixed yields in ECHAM-HAMMOZ, neither for gas-phase chemistry nor for SOA formation itself. The "annual, average mass yields" are now called like this and hopefully clearly understandable.

The section 2.1.2 is somewhat confusing when you discuss the SALSA-VBS system that apparently is not used in the present study but then in Figure 5 you still present results where you use the VBS method.

Reply: The comparison of ECHAM-HAMMOZ to the SALSA-VBS system was done in order to see how two very similar models can differ. The SALSA-VBS system seems to have similar limitations as AeroCom models in terms of SOA formation underestimation. This is the reason why this comparison was added. To clarify this, the introduction in Section 2.1.1 was extended to better guide the reader and reduce confusion.

On Page 10, L12-13 you write: "Loss processes for SOA in HAM-SALSA include sedimentation, deposition and wash out in the aerosol phase." But should it not be "Loss processes for SOA in HAM-SALSA include, apart from evaporation, sedimentation, deposition and wash out in the aerosol phase." I dont understand why/how the pH of the aerosol particles can differ by several order of magnitudes if the only inorganic species is ammonium bisulfate. I guess the pH is only calculated for the inorganic aqueous phase. Is the 4 order of magnitude difference in pH only related to the average RH at different locations? Please clarify this in the text on page 10 and also refer to the figure in the supplement.

Reply: Page 10, L12-13: yes, the evaporation sink is missing in this sentence, it was adjusted accordingly.
The second part of the comment: The aerosol pH calculation is done with strong assumptions and we admit it is a crude estimate for testing the model. We do take into account water taken up by all compounds, but we assume that they don't affect the activity of hydrogen ion and thus pH (the negative logarithm of the hydrogen ion activity). Which means that the difference is related to the average RH at different locations. The text on page 10 was adjusted and the Figure in S2 referenced.

I miss labeling (e.g. a, b, c ) of the panels in all figure and the figure text to figure 2 need to describe what the two panels actually display.

Reply: Labeling of all panels in all figures was added and the figure text of Figure 2 extended.

In the results section on Page 14 and maybe on some other places you use the terms "quite big" and "quite high". Try to avoid using these vague statements and instead write concrete numbers (e.g. contributes to 30 % of the SOA mass instead of contributes to quite big amounts of the SOA mass).

Reply: The vague statements were specified throughout the manuscript.

On Page 14, L10 you state that LSOPOOHOOH-SOA is extremely low volatile but it is not. At least not at 298.15 K (see the comment concerning LVOC above).

Reply: Indeed, the wording is misleading, we wanted to express it is the most low volatile among the compounds in MOZ. The text was changed accordingly.

Concerning the uncertainty of the estimated saturation vapor pressures (Section 3.3). I think it is good that you discuss the uncertainties related to the functional group contribution methods used to estimate the pure liquid saturation vapor pressures.

In publication by Kurtn et al., J. Phys. Chem. A 2016, 120, 2569–2582 it is stated that: "Unfortunately, the basis data sets of existing group contribution based empirical parametrizations for determining saturation vapor pressures of organic molecules (e.g., SIMPOL, or the widely used and generally successful Nannoolal et al. approach) do not contain complex polyhydroperoxides (or multiply substituted peroxy acids), and the parametrizations may therefore not be reliable for HOMs." LSOPOOHOOH can be considered as a HOM with two hydroperoxide functional groups.

Thus, according to the results from Kurtn et al., 2016 it may have been more appropriate to use the EVAPORATION method which also is provided via UManSysProp (Topping et al., 2016). With EVAPORATION/Nannoolal I get a $p_0$* (298.15 K) of 10-5 Pa, e.g. 38 times higher than the value you used at 298.15 K. In fact according to EVAPORATION LSOPOOHOOH is not a LVOC but a SVOC ($C$* $\approx$ 1 $\mu$g m$^{-3}$) at 298.15 K.

Reply: An additional simulation using EVAPORATION instead of Nannoolal et al. was run, evaluated and integrated into the uncertainty discussion. The iSOA budget is influenced by the saturation vapor pressures calculated using EVAPORATION. Annual iSOA production for 2012 is reduced by 12.8 %, the burden is reduced by 16.7 %. A detailed Table was added showing the impact on each iSOA precursor. Nevertheless, this study mainly focuses on surface concentrations, there the $10 - 20$ % reduction is not visible. The main conclusion, the importance of LIEPOX and LISOPOOHOOH in ECHAM-HAMMOZ does not change.

**1.2 Technical corrections**

Write out that ISOP(OOH)2 refers to dihydroxy dihydroperoxide in the abstract

Reply: Changed.

Page 4, L16: Change from "... and its evaluation it is referred ..." to e.g. "... and its evaluation the reader is referred ..."

Reply: Changed.

Page 4, L33: Change from "The semi-explicit isoprene oxidation with 147 reactions constitutes a major of these reactions in JAM3." e.g. to: "Thus, the 147 reactions in the semi-explicit isoprene oxidation scheme constitute a substantial fraction of these reactions in JAM3."

Reply: Changed.

Page 5, L20: Change from "For simplicity there are ..." to "For simplicity they are ..."

Reply: Changed.

On Page 15, L24: I think it should be "lifetime" and not "life time". Try to reformulate the wording "huge wet deposition loss".

Reply: "life time" was changed to "lifetime". The huge wet deposition loss was deleted and we added some sentences on the wet deposition loss clarifying the rather short lifetime of iSOA in ECHAM-HAMMOZ.

On Page 17, L3: Remove "is" after iSOA and remove one "motivated" on L5.

Reply: Removed.

On Page 17, L14: What do you mean with VBS classes 0, 1 and 10? If it is $C^*$ in $\mu g\,m^{-3}$ non of these species are LVOC at 298.15 K according to e.g. Donahue et al., 2013.

Reply: The naming of the VBS classes was misleading and was adjusted. Now they are called: VBSnonvol (former VBS0), VBS0 (former VBS1), VBS1 (former VBS10). The class VBSnonvol is for non volatile oxidation products, which condense and do not evaporate. VBS0 and VBS1 refer to compounds with $\log 10(C^*) = 0$ and $\log 10(C^*) = 1$ respectively. VBS0 and VBS1 are classified as SVOC, according to the Donahue et al. definition. The whole Section 3.2.1 was corrected, to clarify the comparison.

---

## Author Comment (AC3) · 23 May 2018

**1 Anonymous Referee #2**

**1.1 General comments**

In this manuscript, the authors have developed the global chemistry climate model ECHAM-HAMMOZ to include a more explicit coupling between the gas and particle phase models in order to describe the formation isoprene derived secondary organic aerosol (SOA). With their model, they predict that most of the iSOA is produced by IEPOX and $Isop(OOH)_2$. This ultimately leads to over predictions of iSOA in relatively pristine locations where models typically under predict SOA in general. There seems to be a growing tendency in the literature for models to capture ever increasing complexity in the chemical mechanisms because they are more capable of describing the wide variance of atmospheric conditions. For this reason, I believe this manuscript has a lot of value to interested readers. However, it is incumbent on the developers to describe in detail all the relevant additions to the mechanism and justify other aspects that were not considered. I feel that certain aspects of the chemical mechanism were not adequately characterized in this manuscript. For example, there was no discussion on HOx recycling in the mechanism (an important facet in low NOx regimes) or how specifically all the percentage yields in Figure 1 were obtained  both of which affect oxidation state, product yields and branching ratios  and therefore model results. For this reason, I would reconsider this publication after addressing the major and minor revisions detailed below.

Reply: We thank the referee for the positive comments and for the interest in additional details concerning the model chemical mechanism formulation. Indeed, the resulting yields are global, annual averages and Referee #1 criticized this point, as well. It seems, we did not formulate the text clear enough to be easily understandable. This, and all further specific comments, are addressed below.

**1.2 Specific comments**

1. The chemical mechanism as shown in Figure 1 contains many percentage yields. The authors described the reaction pathways and mentioned yields in the text on pages 6 and 7 although they either did not provide references or a brief discussion of how the yields were obtained. It may be stated in another reference but the crucial reaction yields shown in the figure need to be justified. For example, how was the 9 % gas phase yield of LISOPOOHOOH obtained or how was the 1 % gas phase yield of LC578OOH determined? It is these numbers which will directly affect SOA yields and it is therefore crucial to understand their uncertainties based on how they were derived. A discussion with pertinent references should be included on pages 6 and 7.

   Reply: The yields in Figure 1 and in the text are diagnosed after the one year simulation in 2012. This means that globally 9 % of isoprene carbon mass is converted

into LISOPOOHOOH. After LISOPOOHOOH is produced in the gas phase, it partitions into the aerosol, which is a reversible process. Nevertheless, globally, the net yield of LISOPOOHOOH into the aerosol is 79 %. These numbers are derived from ECHAM-HAMMOZ reference simulation, thus the citation is "missing". Indeed, also Referee #1 asked about the yields. The text was adjusted to clarify these yields, as not fixed, but resulting on an global annual mean. Regionally, the yields vary according to chemical regime (radicals present) and the pre-existing aerosol for reactive uptake and partitioning.

2. $HO_x$ recycling remains an issue in atmospheric chemistry models because $HO_x$ levels are typically under predicted in areas of low NOx. [Archibald et al., "Impacts of HOx regeneration and recycling in the oxidation of isoprene: Consequences for the composition of past, present and future atmospheres", Geophys. Res. Letters, 2011, L05804.] Certain reactions will rapidly consume $HO_x$ such as the formation of LISOPOOHOOH (2 OH and 2 $HO_2$ radicals typically consumed) while other reactions will recycle $HO_x$ such as the ring closure reaction of the IsopOOH-(OH)2 radical to form IEPOX or intramolecular hydrogen shift reactions. The consumption of $HO_x$ species has been expressed in the R1-R22 reactions but there seems to be no mention of $HO_x$ species regeneration which affects the oxidative capacity of the atmosphere in regions of low $NO_x$. For example, reaction R3 will release OH radicals when IEPOX is produced but this is not specified in the reaction. A hydrogen shift reaction (not really discussed in any of the reactions as far as I can tell) may produce carbon centered radicals at hydroxyl sites that may react with O2 to yield a carbonyl compound and $HO_2$. These regenerated $HO_x$ species are important and need to be accounted for and/or discussed in the paper in the section describing these reactions.

Reply: The atmospheric chemical mechanism JAM3 includes reactions, which lead to $HO_x$ recycling. Also some of the reactions described in the manuscript recycle OH and $HO_2$, but the radicals were not explicitly mentioned. This was adjusted, now radicals (OH, NO, $HO_2$) produced in reactions are given in the text. Moreover, JAM3 includes the 1,4H-shift of LC587O2 (precursor to LC587OOH), 1,6H-shift of LISOPOACO2 (lumped species ISOPAO2, ISOPOBO2), 1,5H-shift of ISOPBO2 and ISOPDO2, 1,6H-shift of LHC4ACCO3 (lumped species HC4ACO3, HC4CCO3). These H-shifts yield substantial OH either directly or via subsequent oxidation similarly as in the MOM mechanism (Lelieveld, Jos, et al. "Global tropospheric hydroxyl distribution, budget and reactivity." Atmospheric Chemistry and Physics 16.19 (2016): 12477.) and the LIM1 mechanism (Peeters, Jozef, et al. "Hydroxyl radical recycling in isoprene oxidation driven by hydrogen bonding and hydrogen tunneling: The upgraded LIM1 mechanism." The Journal of Physical Chemistry A 118.38 (2014): 8625-8643.).

DOMENICO? More needed here?

3. On page 6 of the manuscript (line 7) it states: "Not included is the H-shift of LISOPOOHO2 that yields much more volatile compounds than LISOPOOHOOH". I do not agree with the authors that the compounds produced would be 'much more volatile' and therefore

are not relevant to particle phase partitioning. If a 1,5-H-shift occurs in LISOPOOHO2, it would lead to a compound similar to LC578OOH except it would be heavier by one oxygen atom (i.e. a hydroxy-dihydroperoxy carbonyl derivative instead of the LC578OOH diol). Because LC578OOH partitions to the particle phase, so too would this newly produced compound derived from an H-atom shift. This product is indeed less volatile than LISOPOOHOOH, but it would be expected to partition into a particle phase thereby decreasing the influence of LISOPOOHOOH in the mechanism.

Reply: The sentence "If a 1,5-H-shift occurs in LISOPOOHO2, it would lead to a compound similar to LC578OOH except it would be heavier by one oxygen atom" motivated us to do two additional simulations taking into account the possible 1,5-H-shift in LISOPOOHO2. From D'Ambro et al. 2017 (DOI: 10.1021/acs.est.7b00460) Supplement Table S1 and Table S2 we took the idea of the possible product and the rate constant. The product of the 1,5-H-shift is a highly functionalized epoxide (SIMLES: CC(C(OO)C=O)(O1)C1O, Table S1). Introducing a new compound into the chemical mechanism requires a lot of adjustments in different parts of the code, especially when the molecule might be undergoing two processes, partitioning and reactive uptake. To keep it simple, two tests (3 month simulation, June, July and August 2012) were run: one introducing 1,5-H-shift in LISOPOOHO2 forming LIEPOX and a second one introducing 1,5-H-shift in LISOPOOHO2 forming LC578OOH. In both sensitivity simulations an unimolecular rate constant of $0.3\ s^{-1}$ is used.

The results of both sensitivity simulations show reductions in iSOA burden, for the simulated time period (JJA), of 30 %. To discuss these additional results, we introduced a new Section called "Uncertainty in LISOPOOHOOH aerosol". The corresponding plots are shown in the Supplement 2, Figures S2 and S3.

4. The product branching ratios for the subsequent reactions of IsopO2 in Figure 1 seem fixed regardless of the environment. Is this the intended assumption? Because all the subsequent reactions of IsopO2 are bimolecular, the branching ratios (and therefore product yields) will depend on the relative concentrations of RO2, HO2 and NO radicals. The gas phase product yields will therefore not only be influenced by local isoprene concentrations but also on the relative concentrations of these radicals. A discussion of this effect should be included in the manuscript along with a justification as to why using these fixed values represents an average isoprene environment.

Reply: The product branching ratios are not fixed and depend on the ambient radical concentrations. Instead, the yields here result from a global, average and indeed represent isoprene oxidation in ECHAM-HAMMOZ. ECHAM-HAMMOZ seems not to resolve high $NO_x$ environments well, this might be caused by the coarse grid resolution of around $200 \times 200$ km. Further analysis of 3 hourly values in different grid boxes with high isoprene emissions and high $NO_x$ showed, that $NO_s$ suppression of iSOA precursors is resolved by the model, but lost once monthly and annual averages are considered.

5. The acronyms used to describe the chemical mechanism are not very clear. For instance, I cannot figure out what IsopOH is. I presume that IsopOOH is a hydroxy-hydroperoxy isoprene species (of which there are 8 isomers) so does that mean IsopOH is the diol? Chemical structures for all species listed in Table 1 and Figure 1 would be extremely useful.

Reply: Indeed, the chemical structures would clarify the names. As ISOPOOH etc. are simplifications within Figure 1 and text, we added a table in the supplemental material 2 which contains all structures, SMILES codes, the names and corresponding names in MCM. MOZ includes many lumped species. Thus, all isomers are shown.

After major revisions to the section describing the MOZ isoprene chemistry, we removed the minor pathway through the diol ISOPOH, therefore it does not appear in the new tables in Supplement 2.

6. In the chemical mechanism, there is no mention of dinitrate formation which is likely to occur in high NO$_x$ environments.[see Piletic et al. "Barrierless Reactions with Loose Transition States Govern the Yields and Lifetimes of Organic Nitrates Derived from Isoprene", J. Phys. Chem. A, 2017, 8306 and Jenkin et al. The MCM v3.3.1 degradation scheme for isoprene, Atmos. Chem. Phys., 2015, 11433.] These species are highly oxidized and relatively heavy and therefore may affect the SOA yield in high NO$_x$ regime.

Reply: JAM3 includes organic nitrates from isoprene (LNISOOH, which is considered as iSOA precursor, LISOPNO3OOH, LISOPNO3NO3 (dinitrate from isoprene), LISOPNO3O2 etc., see tables S3 and S4 in Supplement 2). They are not considered as iSOA precursors here, because their saturation vapor pressure $p_0^*$ at 298.15 K is not lower than 0.01 Pa, when the group contribution method by Nannoolal et al. (2008) is used. Nevertheless, they might affect the SOA yield, but this is not tested within this study, because the large amount of computational resources needed for each new SOA-tracer, which is defined 11 times (1 gas-phase tracer + 10 SALSA aerosol bin tracers).

**1.3 Technical comments**

1. On page 3 line 21, remove "the" for In the light of

Reply: Corrected.

2. On page 5 line 14, replace "., it is referred to the " with ", the reader is referred to the "

Reply: Corrected.

3. On page 5 line 17, replace "Also the O3 initiated reactions pathways are included in MOZ, but none of the products was low volatile enough." with "The O3 initiated reaction pathways are included in MOZ, but the products are too volatile to contribute to SOA."

Reply: Corrected.

4. On page 9 line 17, add a comma between the words "dependence" and "sensitivity".

Reply: Added.

5. On page 9 line 25, replace "'...processes to only the aerosol sized that are relevant..." with "...processes to include only the aerosol sizes that are..."

Reply: Corrected.

6. On page 9 line 30, replace "Here this model..." with "Here, the model..."

Reply: Corrected.

7. On page 12 line 24, replace "... especially LISOPOOHOOH molar mass of 168.14 g/mol is very large." with "...especially due to LISOPOOHOOH which has a molar mass of 168.14 g/mol that is very large."

Reply: Corrected.

8. On page 14 line 2, replace "... 2-methyltetrols in the order of ng/m3 are measured in..." with " 2-methyltetrols are present in ng/m3 concentrations in the..."

Reply: Corrected.

9. On page 14 lines 9 and 18, the sentences are poorly expressed and need to be rewritten (i.e. "On the LISOPOOHOOH-SOA plot..." and "Hodzic et al")

Reply: The sentences were rewritten.

10. On page 14 line 33, the sentence should read "24 % of isoprene ends up as IEPOX, 9 % as LISOPOOHOOH, ..." where every 'in' is replaced with 'as'

Reply: Corrected.

11. On page 15 line 7, it should read "The majority of precursors are destroyed chemically ..."

    Reply: Corrected.

12. On page 15 line 19, replace "... AeroCom mean value, because iSOA..." with "... AeroCom mean value because the iSOA..."

    Reply: Corrected.

13. On page 17 line 5, remove "motivated" (word duplicated)

    Reply: Corrected.

14. On page 17 line 21, replace "in contrary as can be seen in Figure 5 iSOA..." with "On the contrary as can be seen in Figure 5, iSOA..."

    Reply: Corrected.

15. On page 20 line 14, replace "...of the various different organic " with "for the different organic compounds..."

    Reply: Corrected.

16. On page 20 line 17, replace "...using for all of them 30 kJ/mol." with "using 30 kJ/mol as the $\Delta H_{vap}$."

    Reply: Corrected.

17. On page 21 line 11, replace the word 'at' with 'for' at the end of the line.

    Reply: Corrected.

18. On page 21 line 14, switch the order of "holds also" to "also holds".

    Reply: Corrected.

19. On page 22 line 20, remove the word 'atmospheric' (redundant).

    Reply: Removed.

20. On page 22 line 21, add the word 'the' before AMS at the end of the line.

    Reply: Added.

21. On page 25 line 3, replace the word 'pure' with 'purely'.

    Reply: Corrected.

22. On page 25 line 8, the sentence either needs to be split up or more clearly stated.

    Reply: The sentence was split and clarified.

23. On page 26 line 12, add the word 'the' between 'has most'

    Reply: Added.

24. On page 26 line 16, add the word 'by' between 'followed OH'

    Reply: Added.

25. On page 26 line 17, the sentence should read "... $NO_x$ dependent pathways..."

    Reply: Corrected.

---

## Author Response (AR1)

**Manuscript changes in "Isoprene derived secondary organic aerosol in the global aerosol-chemistry-climate model ECHAM6.3.0-HAM2.3-MOZ1.0"**

The manuscript was modified as described in our replies to the two referees and the executive editor. Major changes are listed below and are highlighted in this pdf wit a blue color for new additions and a red crossed-out text for deletions.

1. Two bug fixes slightly changed the results of the reference run. All numbers and figures were updated accordingly. The main results were not changed by these bug fixes and the effects are comparably low.

2. Section 2.1.1 was revised to also include the radical regeneration and clarify the fact, that no fixed chemical yields are used, but the yields described here result as annual, global averages.

3. Additional sensitivity simulations were performed to fulfill the requirements of both reviewers. Therefore, Section 2.2 was updated and a new section 3.2.5 was written.

4. Section 3.2.1 was revised in order to make this comparison better understandable for the readers.

[revised manuscript text omitted]

---

## Author Response (AR2)

**Reply to Topical Editor Decision: Publish subject to minor revisions (review by editor) (03 Jul 2018) by Gerd A. Folberth**

Dear Gerd Folbert,

Thank you very much for the assistance. I am very glad to hear about the acceptance of the manuscript after including the minor revision. Please find the modified version uploaded.

One comment from my side, I think there was a typo in the Reviewer's comment in the sentence: "Even though the products are arguably less volatile than LISOPOOHOOH they may still give rise to substantial SOA." The product described in D'Ambro 2017b is more volatile than LISOPOOHOOH and the word "more" fits well into this sentence. Nevertheless, I clarified the importance of the 1,5H-Shift on pages 6 and 7.

Best wishes,
Scarlet Stadtler

**Reply to "Suggestions for revision or reasons for rejection" by Anonymous Referee #2**

The manuscript has addressed concerns of the gas phase mechanism specifically with regards to product yields and HOx recycling. I recommend the manuscript for publication provided the authors address the following minor comment. On page 7, the authors mention that the 1,5 H-shift of LISOPOOHO2 is not included in the JAM3 mechanism although its potential importance should be commented on further. The work of D'Ambro et al. (2017b in manuscript) suggest that its production rate is higher than 0.1 s-1. The estimated bimolecular rate for a peroxy radical reaction with HO2 is 1.7E-11 cm3/molec*s (Kirchmer et al. J. Geophys. Res. v.101, 1996, pp. 21007-21022). If the approximate concentration of HO2 in the atmosphere is 1E9 molec/cm3 (Lu et al. Atm. Chem. Phys. 2012, v.12, pp.1541-1569) than the production of LISOPOOHOOH has an effective 1st order rate constant of 0.017 s-1. These estimations suggest that the 1,5 H-shift products may be even more prevalent than LISOPOOHOOH. Even though the products are arguably less volatile than LISOPOOHOOH they may still give rise to substantial SOA. For example, the suggested epoxide in D'Ambro et al. may potentially hydrolyze in the particle phase much like IEPOX. Some of this discussion should be included in the appropriate section on page 7.

Reply: Thank you very much for the constructive feedback. We clarified the importance of the 1,5H-shift reaction on pages 6 and 7 by adding some further details and referring to the respective discussion section. There the impact of the 1,5H-shift reaction of LISOPOOHO2 reaction on iSOA formation is tested and discussed. Indeed, the LISOPOOHOOH production is reduced by more than 90 %, while iSOA production changes by around 30 % due to higher

volatility of the 1,5H-shift products. Nevertheless, there is still a substantial iSOA formation from these products.

**List of changes**

All changes are highlighted blue in the manuscript below.

- The text on pages 6 and 7 was clarified according to the minor revision by Referee # 2.

- In the acknowledgments, we added three project numbers.

[revised manuscript text omitted]

$$\text{LISOPOOHOOH} + \text{OH} \rightarrow \text{LC578OOH} + \text{OH} \tag{R8}$$

LC578OOH is a lumped species representing two MCM species C57OOH and C58OOH. LC578OOH is more volatile than LISPOOHOOH and can be formed via another pathway, as well.

$$
\begin{array}{rcll}
\text{ISOPO2} + \text{NO} & \rightarrow & \text{LHC4ACCHO} + \text{NO2} & \text{(R9)} \\
\text{ISOPO2} + \text{NO}_3 & \rightarrow & \text{LHC4ACCHO} + \text{NO2} & \text{(R10)} \\
\text{ISOPNO3} + h\nu & \rightarrow & \text{LHC4ACCHO} + \text{NO2} + \text{HO}_2 & \text{(R11)} \\
\text{ISOPOOH} + h\nu & \rightarrow & \text{LHC4ACCHO} + \text{OH} + \text{HO}_2 & \text{(R12)} \\
\text{LHC4ACCHO} + \text{OH} & \rightarrow & \text{LC578O2} & \text{(R13)} \\
\text{LC578O2} + \text{HO}_2 & \rightarrow & \text{LC578OOH} & \text{(R14)}
\end{array}
$$

Reactions (R9 – R14) show LC578OOH formation via LHC4ACCHO degradation. LHC4ACCHO is a lumped species representing the MCM species HC4ACHO and HC4CCHO. Finally, LHC4ACCHO is oxidized by OH (R13) and forms LC578O2, which reacts with $HO_2$ to LC578OOH (R14). LC578OOH either reacts with OH back to LC578O2 or is photolysed. LC578O2 can undergo an 1,4 H-shift and recycle OH like for one of the RO2 from methacrolein (Crounse et al., 2011). On a global, annual, average for 2012, just 1 % of the oxidation of total isoprene carbon mass leads to LC578OOH.

The third compound formed from the OH initiated oxidation of isoprene is C59OOH. Starting from ISOPO2, there are two possible oxidation ways for C59OOH formation, one with nitrates as intermediates and a second one where nitrogen oxide is not required. The nitrate pathway starts with formation of ISOPNO3 from ISOPO2 (R5) and continues with OH reaction to form isoprene nitrate peroxy radicals ISOPNO3O2 (R15), which is again a lumped specie.

$$
\begin{array}{rcll}
\text{ISOPNO3} + \text{OH} & \rightarrow & \text{ISOPNO3O2} & \text{(R15)} \\
\text{ISOPNO3O2} + \text{CH3O2} & \rightarrow & \text{ISOPNO3OOH} & \text{(R16)} \\
\text{ISOPNO3OOH} + \text{OH} & \rightarrow & \text{C59OOH} & \text{(R17)}
\end{array}
$$

[revised manuscript text omitted]